

# Black hole mirages: Electron lensing and Berry curvature effects in inhomogeneously tilted Weyl semimetals

Andreas Haller[1,*,†], Suraj Hegde[2,†], Chen Xu[2],
Christophe De Beule[1,3], Thomas L. Schmidt[1] and Tobias Meng[2]

**1** Department of Physics and Materials Science, University of Luxembourg,
L-1511 Luxembourg, Luxembourg
**2** Institute of Theoretical Physics and Würzburg-Dresden Cluster of Excellence ct.qmat,
Technische Universität Dresden, 01069 Dresden, Germany.
**3** Department of Physics and Astronomy, University of Pennsylvania,
Philadelphia PA 19104, USA.

* andreas.haller@uni.lu

## Abstract

We study electronic transport in Weyl semimetals with spatially varying nodal tilt profiles. We find that the flow of electrons can be guided precisely by judiciously chosen tilt profiles. In a broad regime of parameters, we show that electron flow is described well by semiclassical equations of motion similar to the ones governing gravitational attraction. This analogy provides a physically transparent tool for designing tiltronic devices like electronic lenses. The analogy to gravity circumvents the notoriously difficult full-fledged description of inhomogeneous solids. A comparison to microscopic lattice simulations shows that it is only valid for trajectories sufficiently far from analogue black holes. We finally comment on the Berry curvature-driven transverse motion and relate the latter to spin precession physics.



## Contents

---

† These authors contributed equally to this work.

# 1  Introduction

Over the last two decades, the combination of solid state and relativistic physics has proven to be a particularly fruitful direction of research. The underlying key observation is that low-energy states in entire classes of solids can be described by variants of the Dirac equation, which had initially been developed for relativistic quantum mechanics. This principle has most prominently been exemplified by the $(2 + 1)$-dimensional material graphene [1]. In recent years, a similar analogy has been identified for certain $(3 + 1)$-dimensional materials, now known as Dirac and Weyl semimetals [2].

Despite connecting two extremely different physical situations, this mapping between equations implies that certain phenomena known from high-energy physics are mirrored in solids, and vice-versa. This includes the analogues of Klein tunneling in graphene [3] and of the chiral anomaly in Weyl semimetals [4–10], as well as a connection between gravitational anomalies and transport in solids [11–14].

Even more recently, it has been proposed that Weyl semimetals with spatially inhomogeneous nodal tilts can be mapped to spacetimes with black holes [15–28]. On a pictorial level, this mapping is one between tilted Weyl nodes in solids, and tilted lightcones in spacetimes with black holes. More formally, the mapping is based on an identification of effective tetrads in inhomogeneous Weyl Hamiltonians. An alternative avenue for wavepacket dynamics mimicking gravitational motion are tight-binding hopping amplitudes with specific spatial structure [29–31]. These studies are part of a broader stream of works connecting gravitational physics with, for example, condensed-matter systems or ultracold quantum gases [32–40].

Using the mapping between low-energy Hamiltonians and Dirac equations, it has been proposed that various types of effective spacetimes can be engineered in solids with inhomogeneous tilts [18, 23]. By extension, it has been reasoned that one might be able to engineer electronic lensing devices that mimic gravitational lensing in suitably designed analogue spacetimes [18, 19, 41], in particular ones with analogues of black holes. At the same time, it has

also been found that the analogy to black hole spacetimes ceases to describe the full physics when the wavepacket reaches the analogue event horizon. As discussed in [26, 27], additional states at low energies but large momenta appear close to the event horizon, and these states have been identified as being crucial for transport. This begs the question about the conditions that apply when using the analogy between inhomogeneous Weyl semimetals and analogue spacetimes, especially in connection to tiltronic devices such as electronic lenses. In the present study, we provide a detailed analysis of electron lensing in Weyl semimetals with inhomogeneous tilts. In particular, we discuss physics that goes beyond the mapping between semiclassical trajectories obtained for a linearized continuum model, and geodesics.

First, the dispersion of any lattice model with Weyl nodes necessarily ceases to be linear at higher energies. By taking into account model-specific nonlinearities of the dispersion near the Weyl nodes, we show that wavepacket trajectories in solids exhibit important deviations from the geodesics that the naive mapping between inhomogeneously tilted Weyl semimetals and black hole spacetimes translates them to. Electrons can for example enter the region within the analogue event horizon and leave again.

Second, we analyze a fully microscopic tight-binding model that includes not only band curvature effects, but also more than one Weyl node, and interfaces with leads. These simulations show that electron trajectories exhibit a characteristic bending analogue to gravitational lensing when they are sufficiently far from the overtitled region forming the black hole analogue. When they approach this region, the trajectories exhibit a range of fascinating behaviors, including for example slingshot trajectories. Intriguingly, we can still connect this behavior to, for example, the presence of an analogue photon sphere in the semiclassical equations of motion.

Overall, we thus find the analogy between overtilted Weyl nodes and black holes to work best for wavepackets far from the overtilted region. The more closely the black hole analogue region is probed, the less it behaves like a black hole. We therefore dub the overtilted region a black hole mirage.

We organized our analysis as follows. In section 2, we review the semiclassical equations of motion describing electron transport in Weyl semimetals. Section 3 analyzes these equations of motion for a cylindrical tilt profile, and identifies the electronic analogues of lensed and captured trajectories of light close to black holes, as well as a generalized counterpart of the photon sphere in a system with axial symmetry. The impact of lattice physics is commented on in section 4, both on the level of using the lattice dispersion in the semiclassics, and for the fully microscopic tight-binding analysis. We conclude the analysis in section 5 by discussing electronic motion parallel to the cylinder axis, which is governed by Berry curvature physics, and connect this transverse motion to the dynamics of the electron spin. We finally conclude with an outlook on experimental prospects in section 6.

## 2 Weyl semimetals with spatially varying tilt profiles: review of semiclassical dynamics

To set the notation and make the paper self-contained, we begin by reviewing the semiclassical equations of motions for Weyl semimetals. Physically, we consider a (3+1)-dimensional material whose band structure exhibits a number of Weyl nodes close to the Fermi level. We furthermore assume that there are no additional trivial electron or hole pockets present. The low-energy Hamiltonian is then a collection of Weyl Hamiltonians, one for each node. Measuring the momentum $\boldsymbol{k}$ as well as the energy relative to the respective node, those read

$$H_1(\boldsymbol{k}) = (\boldsymbol{u} \cdot \boldsymbol{k}) \sigma_0 + \chi \, v_F \, \boldsymbol{k} \cdot \boldsymbol{\sigma} \,, \tag{1}$$

where $v_F$ is the Fermi velocity, $\chi = \pm 1$ is the node's chirality, $\sigma_0$ denotes the $2 \times 2$ identity matrix, and $\boldsymbol{\sigma}$ is the vector of Pauli matrices.

Our model also features a nodal tilt characterized by $\boldsymbol{u}$. More precisely, eq. (1) describes a Weyl node with a spatially constant tilt. In the remainder, we will be interested in spatially varying tilt profiles, $\boldsymbol{u} \to \boldsymbol{u}(\boldsymbol{r})$ (and therefore $H_1(\boldsymbol{k}) \to H_1(\boldsymbol{r}, \boldsymbol{k})$). For a general tilt profile, states with arbitrary momentum difference get coupled. We, however, focus on tilt profiles that are smooth on the relevant electronic length scales. Those particularly include the scales set by the inverse momentum distance between Weyl nodes, and the inverse of the implicitly present momentum cutoff for linearization in eq. (1) (this is addressed in more detail in section 4). In this case, we can use the linearized Hamiltonian for individual nodes with the Hermitian replacement [26, 27]

$$\boldsymbol{u} \cdot \boldsymbol{k} \to \boldsymbol{u}(\boldsymbol{r}) \cdot (-i\boldsymbol{\nabla}) + \frac{1}{2}(-i\boldsymbol{\nabla} \cdot \boldsymbol{u}(\boldsymbol{r})) \,. \tag{2}$$

Such a Hamiltonian models a well-defined nodal band structure with a locally varying tilt. We therefore employ the same semiclassical equations of motion as used for wavepackets narrowly peaked in position and momentum space for constant tilts [42], but replace $\boldsymbol{u} \to \boldsymbol{u}(\boldsymbol{r})$. Note that we here also assume $\boldsymbol{u}(\boldsymbol{r})$ to vary on scales much larger than the width of the wavepacket, such that corrections to the semiclassical equations involving the quantum metric can be ignored [43, 44]. This replacement for example means that the eigenenergies are $E_\pm(\boldsymbol{r}, \boldsymbol{k}) = \boldsymbol{u}(\boldsymbol{r}) \cdot \boldsymbol{k} \pm v_F|\boldsymbol{k}|$. For $|\boldsymbol{u}(\boldsymbol{r})|/v_F < 1$ ($|\boldsymbol{u}(\boldsymbol{r})|/v_F > 1$), the system locally behaves like a Weyl semimetal of type I (type II).

For concreteness, we focus on wavepackets built from states in the $E_+(\boldsymbol{r}, \boldsymbol{k})$-band in the remainder of the paper. In the absence of external electromagnetic fields, those are governed by the semiclassical equations of motion

$$\begin{aligned} \dot{\boldsymbol{r}} &= \boldsymbol{\nabla}_k E_+(\boldsymbol{r}, \boldsymbol{k}) + \dot{\boldsymbol{k}} \times \boldsymbol{\Omega}(\boldsymbol{k}) = \boldsymbol{u}(\boldsymbol{r}) + v_F \hat{\boldsymbol{k}} + \dot{\boldsymbol{k}} \times \boldsymbol{\Omega}(\boldsymbol{k}), \\ \dot{\boldsymbol{k}} &= -\boldsymbol{\nabla}_r E_+(\boldsymbol{r}, \boldsymbol{k}) = -\boldsymbol{\nabla}_r(\boldsymbol{u}(\boldsymbol{r}) \cdot \boldsymbol{k}), \end{aligned} \tag{3}$$

where $\boldsymbol{\Omega}(\boldsymbol{k}) = \chi \boldsymbol{k}/(2k^3)$ is the momentum-space Berry curvature and $k = |\boldsymbol{k}|$. Note that the tilt term is proportional to the identity matrix and therefore does not affect the eigenstates, nor the momentum-space Berry curvature, and there are no real-space Berry curvature contributions [42].

Specific inhomogeneous tilt profiles with spherical symmetry have recently been argued to realize analogues of spacetimes with black holes [17–27, 29]. The connection between the two fields also entails that semiclassical trajectories of wavepackets in inhomogeneously tilted Weyl semimetals can be mapped to geodesics of the corresponding curved spacetime (see appendix A).

Realizing the analogue of an astrophysical black hole requires a rotationally symmetric tilt profile. Here, we aim at exploring transport in such systems, and want to analyze to which degree the analogy with curved spacetimes can be used to describe this transport. A spherically symmetric strain profile, however, seems rather hard to realize in a three-dimensional solid. A more realistic scenario is tilt profiles with axial symmetry, which could in principle result from the strain pattern close to a hole drilled through the material. A microscopic displacement field $\boldsymbol{h}$ can for example give rise to nodal tilts $\boldsymbol{u}$ of the form $u_i \propto (\partial_i h_j) k_{0j}$, where $2\boldsymbol{k}_0$ is the Weyl node separation [45].

We therefore from now on specialize to tilt profiles with axial symmetry, i.e., $\boldsymbol{u}(\boldsymbol{r}) = u(\rho)\hat{\boldsymbol{\rho}}$. Here and below, hats denote unit vectors, $\hat{\boldsymbol{x}} = \boldsymbol{x}/|\boldsymbol{x}|$, $\rho$ is the radius in cylindrical coordinates

$(\rho, \phi, z)$, and $\boldsymbol{\rho} = (x, y, 0)^T$. The semiclassical equations are then given by

$$\dot{\boldsymbol{r}} = \boldsymbol{u}(\boldsymbol{\rho}) + v_F \,\hat{\boldsymbol{k}} + \dot{\boldsymbol{k}} \times \boldsymbol{\Omega}, \tag{4}$$

$$\dot{\boldsymbol{k}} = -\frac{\partial u}{\partial \rho} (\boldsymbol{k} \cdot \hat{\boldsymbol{\rho}}) \hat{\boldsymbol{\rho}} - \frac{u L_z}{\rho^2} \hat{\boldsymbol{\phi}}, \tag{5}$$

where $\boldsymbol{L} = \boldsymbol{r} \times \boldsymbol{k}$ with $L_z = \rho \, \boldsymbol{k} \cdot \hat{\boldsymbol{\phi}}$ is the orbital angular momentum. By combining the above equations, we obtain

$$\dot{\boldsymbol{r}} = \boldsymbol{u}(\boldsymbol{\rho}) + v_F \,\hat{\boldsymbol{k}} - \frac{\chi}{2k^3} \left[ \frac{\boldsymbol{k} \cdot \hat{\boldsymbol{\rho}}}{\rho} \left( \frac{\partial u}{\partial \rho} - \frac{u}{\rho} \right) L_z \,\hat{\boldsymbol{z}} + k_z \left( \frac{u L_z}{\rho^2} \hat{\boldsymbol{\rho}} - (\boldsymbol{k} \cdot \hat{\boldsymbol{\rho}}) \frac{\partial u}{\partial \rho} \hat{\boldsymbol{\phi}} \right) \right]. \tag{6}$$

This highlights the two contributions to the motion of semiclassical wavepackets. The first and second terms combined correspond to the group velocity $\boldsymbol{\nabla}_k E_+(\boldsymbol{r}, \boldsymbol{k})$, which depends on position via the tilt. The remaining terms, proportional to $\chi$, derive from the Berry curvature. This geometric contribution to the wavepacket dynamics is also known as the anomalous velocity.

It is finally helpful to note that the semiclassical equations of motion entail several conserved quantities. First, $\dot{E}_\pm(\boldsymbol{r}, \boldsymbol{k}) = 0$, physically resulting from the Hamiltonian being time-independent. Second, the rotational symmetry around the $z$-axis implies that $\dot{J}_z = 0$, where $\boldsymbol{J} = \boldsymbol{L} + \boldsymbol{s}$ is the total angular momentum, and $\boldsymbol{s} = \chi \,\hat{\boldsymbol{k}}/2$ is the spin, defined by the expectation value of the spin operator $\boldsymbol{\sigma}$ of a state with momentum $\boldsymbol{k}$ in the $E_+$-band as $s_i = 1/2 \langle \sigma_i \rangle$. Third, translation invariance along $z$ entails $\dot{k}_z = 0$. Lastly, the equations of motion are invariant after a combined transformation of $u \to -u$, $\boldsymbol{k} \to -\boldsymbol{k}$ and $t \to -t$.

## 3 Cylindrical tilt profiles and perpendicular injection: lensing, capture, and separatrix of trajectories

In the following, we consider general axial symmetric tilt profiles $\boldsymbol{u}(\boldsymbol{\rho}) = u(\rho) \hat{\boldsymbol{\rho}}$ of the form $u(\rho) = -v_F (\rho_t/\rho)^\alpha$. The radius $\rho = \rho_t$ at which $u(\rho) = -v_F$ marks the analogue of an event horizon between a type I and a type II region, while the parameter $\alpha$ determines how fast the tilt decays outside the event horizon analogue [17, 20, 26, 27]. We furthermore focus on incoming wavepackets with initial $k_z = 0$, i.e., injected perpendicular to the cylindrical axis. In this case, $s_z = 0$ and thus $\dot{s}_z = 0$, which together with $\dot{J}_z = 0$ entails $\dot{L}_z = 0$, and the equations of motion along $\rho$ and $\phi$ are decoupled from the one along $z$.

For such a tilt profile, trajectories described by the semiclassical equations of motion in eq. (3) can be categorized into two distinct classes, lensing and capture. A lensed trajectory begins far from the region with strong tilt, approaches this region (but never too closely), and then escapes to infinity. A captured trajectory instead falls into the region with strong tilt; the fate of such a wavepacket propagating in a lattice model is commented on in section 4. These two regimes are separated by a specific trajectory denoted as the "separatrix" that orbits the region of strong tilt.

We begin by noting that the distinction between lensing and capturing is defined by the motion in the $(\rho, \phi)$-plane, onto which we focus in this section. The motion along $z$, governed solely by Berry curvature effects for $k_z = 0$, is commented on in section 5.

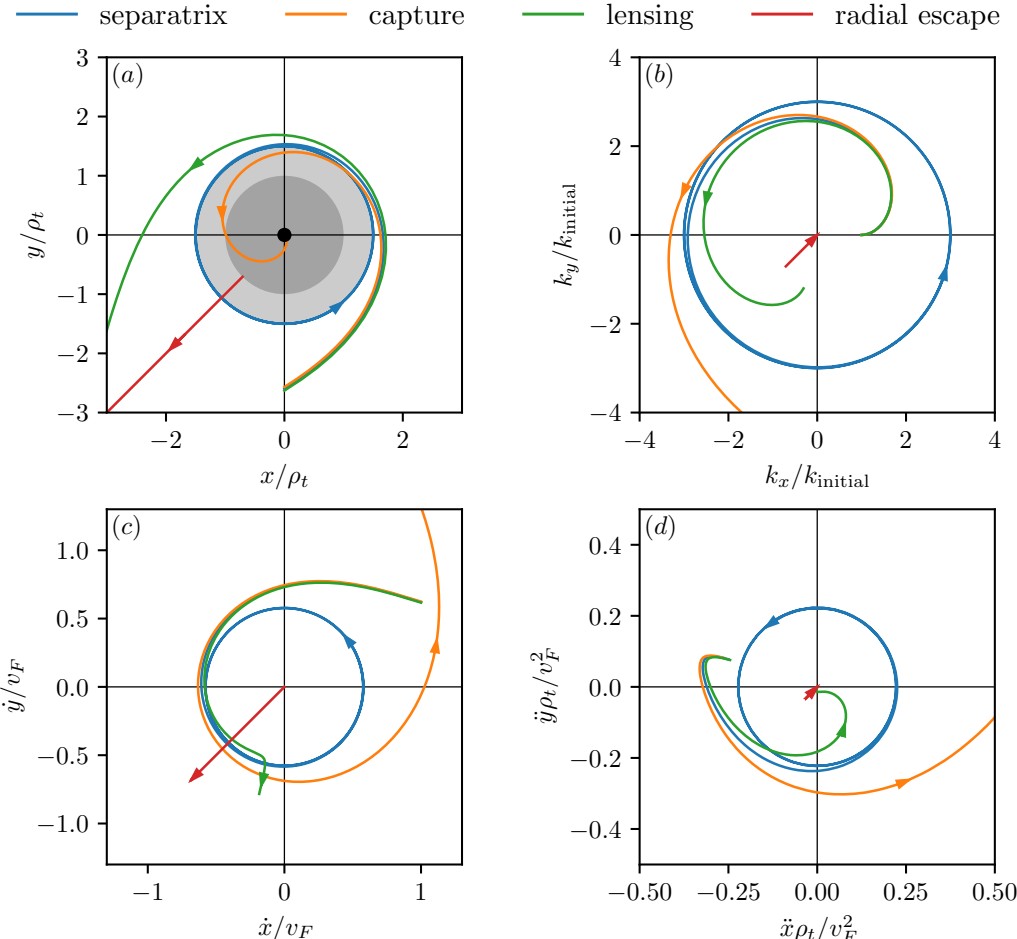

Figure 1: In panel $(a)$ and $(b)$, we display the trajectories given by the solutions of $\boldsymbol{r}(t)$ and $\boldsymbol{k}(t)$ for the tilt profile $u = -v_F \sqrt{\rho_t/\rho}$. Panel $(c)$ shows the velocity components. Only positive evolution times are considered (the directed trajectories are depicted by arrows). The radial escape trajectory starts at a distance $1.0001\rho_t$ from the tilt center, and initial momenta are chosen as $k_x = k_y$, $k_z = 0$. In $(d)$, we display more details on the curvature of the real-space trajectory.

In the $(\rho, \phi)$-plane, the equations of motion read

$$\dot{\rho} = u(\rho) + v_F \frac{\boldsymbol{k} \cdot \hat{\boldsymbol{\rho}}}{k}, \tag{7a}$$

$$\rho \, \dot{\phi} = v_F \frac{\boldsymbol{k} \cdot \hat{\boldsymbol{\phi}}}{k}, \tag{7b}$$

$$\frac{d(\boldsymbol{k} \cdot \hat{\boldsymbol{\rho}})}{dt} - (\boldsymbol{k} \cdot \hat{\boldsymbol{\phi}}) \, \dot{\phi} = -\frac{\partial u}{\partial \rho} (\boldsymbol{k} \cdot \hat{\boldsymbol{\rho}}), \tag{7c}$$

$$\frac{d(\boldsymbol{k} \cdot \hat{\boldsymbol{\phi}})}{dt} + (\boldsymbol{k} \cdot \hat{\boldsymbol{\rho}}) \, \dot{\phi} = -\frac{u(\rho)}{\rho} (\boldsymbol{k} \cdot \hat{\boldsymbol{\phi}}). \tag{7d}$$

Using $k(\rho)^2 = (\boldsymbol{k} \cdot \hat{\boldsymbol{\rho}})^2 + L_z^2/\rho^2$ (for $k_z = 0$) and the dispersion, we express $\dot{\rho}$ as function of the conserved energy $E_+$, the conserved angular momentum $L_z$, and $\rho$. After some

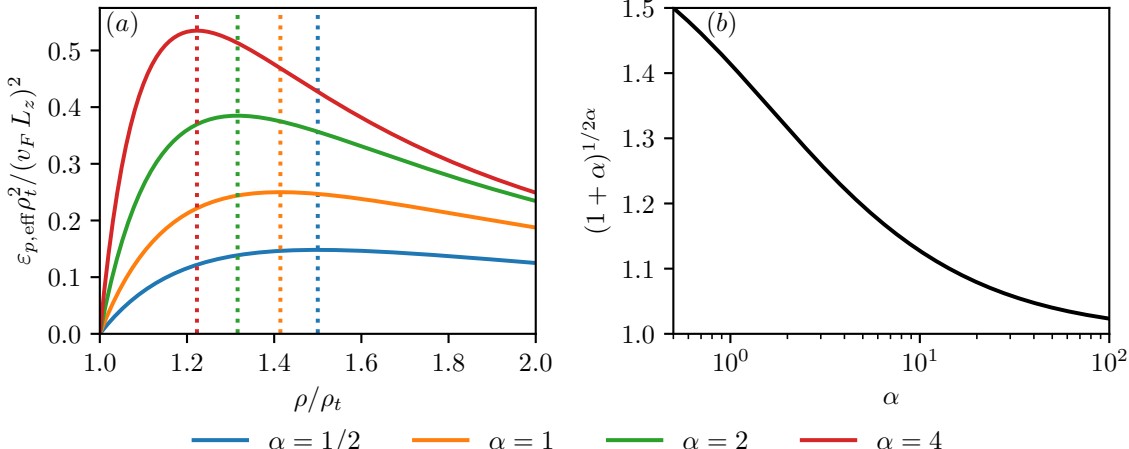

Figure 2: Panel $(a)$ displays the dimensionless effective potential for tilt profiles $u(\rho) = -v_F(\rho_t/\rho)^\alpha$ with different $\alpha$. For all potentials, the analog horizon is located at $\rho = \rho_t$. The maxima of the effective potential (the separatrix radii) are located at $\rho_s = \rho_t(1+\alpha)^{1/2\alpha}$, which we visualize in $(b)$.

manipulations, we find

$$\dot{\rho}^2 = \frac{1}{k^2}\left[E_+^2 - \varepsilon_{\mathrm{p,eff}}(\rho)\right], \tag{8}$$

$$\varepsilon_{\mathrm{p,eff}}(\rho) = \frac{L_z^2}{\rho^2}\left(v_F^2 - u(\rho)^2\right). \tag{9}$$

The radial equation of motion in eq. (8) thus takes the form of an equation of effective energy conservation for a classical free particle with a conserved effective total energy $\varepsilon_{\mathrm{tot,eff}} = E_+^2$ composed of an effective kinetic energy $\varepsilon_{\mathrm{k,eff}} = m_{\mathrm{eff}}\dot{\rho}^2/2$ with an effective mass $m_{\mathrm{eff}} = 2k^2$, and an effective potential $\varepsilon_{\mathrm{p,eff}}(\rho)$. Note that this analogy is not perfect because $k$ is not a conserved quantity. It is sufficient, however, to make the distinction between captured and lensed trajectories: this distinction is based on $\dot{\rho}$ vanishing, or not.

For a general $u(\rho)$, the effective potential $\varepsilon_{\mathrm{p,eff}}(\rho)$ can have both minima and maxima. In case of a Schwarzschild-like tilt profile $u(\rho) = -v_F\sqrt{\rho_t/\rho}$, note that for Weyl fermions ($m = 0$), the standard Newtonian term $-m\rho_t v_F^2/(2\rho)$ is vanishing and only the repulsive potential energy $\propto 1/\rho^2$ and the attractive cubic term contribute to the effective potential [46]. For the general tilt profiles $u(\rho) = -v_F(\rho_t/\rho)^\alpha$ considered here, we find that $\varepsilon_{\mathrm{p,eff}}(\rho)$ has exactly one maximum, see fig. 2. This physically transparent picture explains that depending on the total energy $E_+$ and the maximal value of the effective potential $\varepsilon_{\mathrm{p,eff}}^{\max}$, the trajectory of a particle moving towards the origin can be of three possible types.

1. Lensing: for $E_+^2 < \varepsilon_{\mathrm{p,eff}}^{\max}$, the wavepacket approaches the origin until it reaches its periapsis (the point of closest approach to the strongly tilted region) at $\rho_p$, set by $E_+^2 = \varepsilon_{\mathrm{p,eff}}(\rho_p)$. There, $\dot{\rho} = 0$, which allows to express $\rho_p$ in terms of conserved quantities via the implicit equation $E_+\rho_p = |L_z|\sqrt{v_F^2 - u(\rho_p)^2}$. The wavepacket is then deflected to infinity.

2. Capture: for $E_+^2 > \varepsilon_{\mathrm{p,eff}}^{\max}$, $\dot{\rho}$ never vanishes. For initial negative $\dot{\rho}$, the semiclassical equations then predict the wavepacket continues falling towards the origin.

3. Separatrix: for $E_+^2 = \varepsilon_{\mathrm{p,eff}}^{\max}$, the wavepacket approaches the origin until it reaches the top of the effective potential, where it remains. As we will show below, the wavepacket then

still performs an angular motion, and is thus bound in an (unstable) orbit around the origin, at $\rho = \rho_s = \rho_t(1+\alpha)^{1/2\alpha}$ (see fig. 2).

Note that the separatrix marks a critical radius of closest approach for inward-moving wavepackets (see also discussion in section 3.3 below). Emission of an outward-moving wavepacket is possible for all initial radii outside the analogue event horizon located at $\rho = \rho_t$. As shown by eq. (7a), our choice of tilt profiles entails $\dot{\rho} < 0$ for $\rho < \rho_t$ as then $u < -v_F$. For an initial radius $\rho_{\text{initial}} > \rho_t$, wavepackets can escape at least radially: if $\boldsymbol{k} \cdot \hat{\boldsymbol{\rho}} > 0$ and $\boldsymbol{k} \cdot \hat{\boldsymbol{\phi}} = 0$, we find that eq. (7a) implies $\dot{\rho} = u(\rho) + v_F > 0$, $\ddot{\rho} = u'(\rho)\dot{\rho} > 0$ and eq. (7b) leads to $\dot{\phi} = 0$. This corresponds to an escape of the wavepacket along a straight radial trajectory (see fig. 1 $(a)$). We want to stress here that the radial acceleration away from the origin is entirely caused by the decreasing $m_{\text{eff}}(t) = 2k(0)^2 \exp\left(-2\int_0^t u'(\rho(\tau))d\tau\right)$, which can be eliminated from the equations of motion by using an affine parametrization instead of a temporal one (for more details, see appendix A).

To further describe the two-dimensional motion in the $(\rho, \phi)$-plane, it is convenient to combine the equations for $\dot{\phi}$ and $\dot{\rho}$. Using that $\boldsymbol{k} \cdot \hat{\boldsymbol{\phi}} = L_z/\rho$, we find

$$\frac{d\phi}{d\rho} = \frac{\dot{\phi}}{\dot{\rho}} = \pm \frac{v_F L_z}{\rho^2 \sqrt{E_+^2 - \varepsilon_{\text{p,eff}}(\rho)}}, \tag{10}$$

where the sign corresponds to the sign of $\dot{\rho}$.

## 3.1 Lensing and deflection angle

For $E_+^2 < \varepsilon_{\text{p,eff}}^{\text{max}}$, when the wavepacket first approaches the origin and then escapes to infinity, eq. (10) predicts the asymptotic outgoing motion of the wavepacket to exhibit a deflection angle as compared to the asymptotic incoming motion, see fig. 3. This deflection angle is the key ingredient that allows to build electronic lensing devices.

Using eq. (10), the deflection angle can be calculated very much in the spirit of lensing around black holes [47]. As detailed in appendix B, we find

$$\pi + \varphi = 2 \int_{\rho_p}^{\infty} \frac{v_F \, d\rho}{\sqrt{\frac{1}{\rho_p^2}\left(v_F^2 - u(\rho_p)^2\right)\rho^4 - \rho^2\left(v_F^2 - u(\rho)^2\right)}}, \tag{11}$$

where $\varphi$ is the angle of deflection. In fig. 3, we show that numerical evaluation of the integral in eq. (11) coincides with the deflection angles of trajectories which we obtained by numerical solutions of eq. (3) with different initial conditions. Analytical progress can be made for lensed trajectories with periapsides sufficiently far from the tilt center such that $\rho_p/\rho_t \gg 1$. The calculation detailed in appendix B shows that the deflection angle to leading order in $\rho_t/\rho_p$ is then given by

$$\varphi \approx \frac{u(\rho_p)^2}{v_F^2} \frac{\sqrt{\pi}\,\Gamma\left(\frac{3}{2} + \alpha\right)}{\Gamma(1+\alpha)}. \tag{12}$$

For the concrete example of the Schwarzschild-like tilt profile with $\alpha = 1/2$, i.e., $u(\rho) = -v_F\sqrt{\rho_t/\rho}$, this implies $\varphi \approx 2u(\rho_p)^2/v_F^2$.

## 3.2 Capture

For a trajectory starting outside the separatrix with $E_+^2 > \varepsilon_{\text{p,eff}}^{\text{max}}$ and $\dot{\rho} < 0$ initially, the semiclassical equations of motion predict the wavepacket to always fall towards the origin. Similarly, any trajectory with $E_+^2 < \varepsilon_{\text{p,eff}}^{\text{max}}$ starting inside the separatrix can only remain inside, and spirals

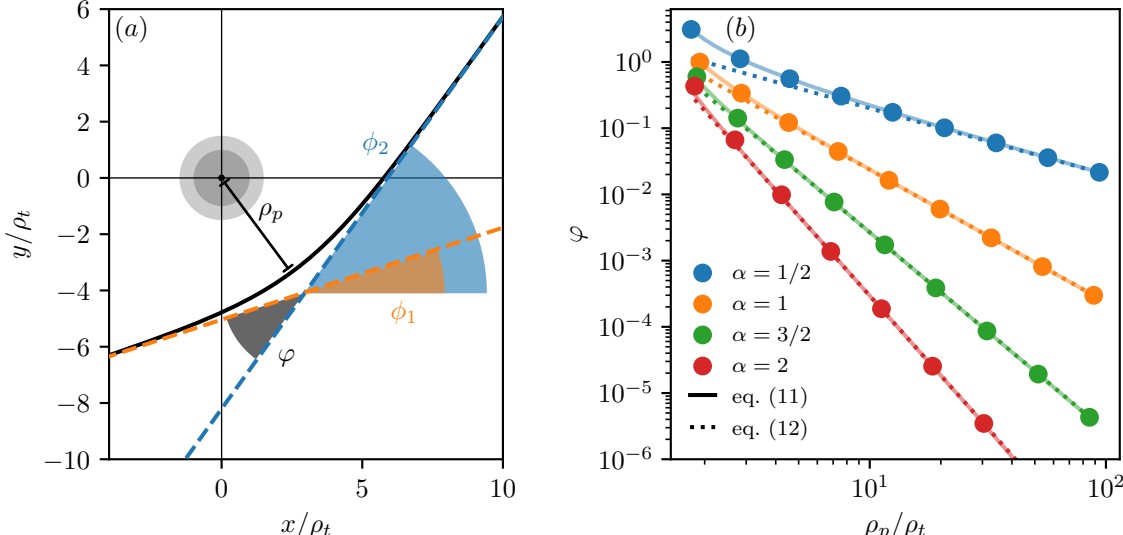

Figure 3: In panel ($a$), we depict a lensing trajectory (black line), which is characterized by two linear segments (blue and yellow dashed), which define the azimuth angles contributing to the deflection angle $\varphi = \phi_2 - \phi_1$. The deflection angle resulting from different periapsides and tilt profiles is presented in panel ($b$). A comparison between numerical integration of eq. (11) and the approximate analytic result in eq. (12) is indicated by solid and dotted lines, respectively. Both equations are contrasted with extracted angles and periapsides from trajectories obtained by numerically integrating the semiclassical equations of motion, eq. (3), for different initial conditions (disks). The color distinguishes between different tilt profile decays $\alpha$.

inward to the origin. During this plunge towards the center, the velocity and acceleration diverge, and the equations of motion in eq. (7) break down. It has for example been discussed in Refs. [26, 27] that additional electronic states at low energies but large momenta appear in lattice systems which have to be taken into account for a faithful description of transport. We postpone the further discussion of trajectories in the capturing regime to section 4.

### 3.3 Separatrix

If $E_+^2 = \varepsilon_{\mathrm{p,eff}}^{\max}$, wavepackets starting outside the separatrix approach the origin up to the separatrix at $\rho_s = \rho_t (1+\alpha)^{1/2\alpha}$, and then remain at that distance. This is further shown by the radial acceleration, which then reads

$$\ddot{\rho}|_{\rho=\rho_s} = -\frac{1}{k^2}\left(\frac{1}{2}\frac{\partial \varepsilon_{\mathrm{p,eff}}}{\partial \rho} + \dot{\rho}\,\dot{k}\,k\right)_{\rho=\rho_s} = 0\,. \tag{13}$$

Using eq. (8) and the fact that $\varepsilon_{\mathrm{p,eff}}(\rho)$ has a maximum on the separatrix, we find that both the radial velocity $\dot{\rho}$ and the radial acceleration $\ddot{\rho}$ vanishes. The wavepacket then orbits the origin at the radius $\rho_s$. However, since the effective potential has a maximum at $\rho_s$, the orbit is unstable. Any perturbation will make the wavepacket either plunge towards the center or escape to infinity.

The semiclassical equations of motion for a spherically symmetric tilt profile are obtained by replacing $u(\rho)\hat{\boldsymbol{\rho}} \rightarrow u(r)\hat{\boldsymbol{r}} = -v_F \sqrt{r_H/r}\,\hat{\boldsymbol{r}}$. They are equivalent to the equations of motion for Weyl fermions in a Schwarzschild spacetime with an event horizon at $r_H$ (we show this, neglecting anomalous velocity terms, in appendix A). In this context, the separatrix does not

lie on a cylinder, but on a sphere, known as the photon sphere. The latter is a set of unstable bound orbits which bifurcate the rest of the geodesic solutions into purely lensing or captured trajectories around a black hole. It has been shown to be a fundamental feature of black hole spacetimes, and it is an active topic of investigation to this day, both in theoretical [48–50] as we all as gravitational wave detection measurements [51]. Experimentally, the photon sphere is the key visible feature in the recent black hole images taken through the Event Horizon Telescope [52], making it one of the few directly observed aspects of black holes. Our analysis generalizes the photon sphere of cosmological black holes to the more general concept of a separatrix between lensed and captured trajectories, here applied to black hole analogues realized in Weyl semimetals with inhomogeneous nodal tilts. Since the separatrix has the shape of a cylinder for tilt profiles with axial symmetry, we call it the analog photon cylinder.

## 4 Lattice simulations: validity of semiclassics, and fate of wavepackets inside the separatrix

In this section we perform large-scale microscopic computations to check the regime of validity of the semiclassical theory presented in section 3, and to understand the fate of wavepackets crossing the separatrix. For computational reasons, we focus on an effective two-dimensional lattice model in terms of a hybrid real and momentum space representation of the Hamiltonian. This implies that possible transverse deflections from Berry curvature contributions cannot be analyzed in a straightforward way.

### 4.1 Effective two-dimensional lattice model

Our starting point is a minimal model for a tilted Weyl semimetal with two Weyl cones. The low-energy continuum form of the Hamiltonian is given by $\hat{H} = \sum_{\mathbf{k}} \hat{c}_{\mathbf{k}}^{\dagger} H_2(\mathbf{k}) \hat{c}_{\mathbf{k}}$ with a two-component annihilation operator $\hat{c}_{\mathbf{k}} = (\hat{c}_{\uparrow,\mathbf{k}}, \hat{c}_{\downarrow,\mathbf{k}})^T$ and a Hamiltonian matrix,

$$H_2(\mathbf{k}) = \left[ u_x k_x + u_y k_y + u_z(k_z + k_0) \right] \sigma_0 + v_F k_x \sigma_x + v_F k_y \sigma_y + \frac{1}{2m^*}(k^2 - k_0^2)\sigma_z. \quad (14)$$

The parameter $k_0 > 0$ separates the two Weyl nodes, one located at $(k_x = 0, k_y = 0, k_z = -k_0)$ and zero energy, the other at $(0, 0, k_0)$ at energy $2u_z k_0$. The additional $1/2m^*$ term gives rise to curvature along the $z$ direction.

In the following discussion, we regularize the continuum model on a square lattice with lattice constant $a$. In order to discuss the trajectories of electrons with momenta close to a single Weyl node, from now on we set $k_z = -k_0$. This is justified because $k_z$ is conserved by translational invariance along the $z$ direction for an axially symmetric tilt profile. We can then focus on a two-dimensional lattice model that coincides with the low energy Hamiltonian in eq. (14) for wave numbers $k_x, k_y \ll 1/a$, namely

$$H_{\text{latt}}(\mathbf{k}) = a^{-1} \left[ (u_x \sin(k_x a) + u_y \sin(k_y a))\sigma_0 + v_F \sin(k_x a)\sigma_x + v_F \sin(k_y a)\sigma_y \right]$$
$$+ \frac{1}{m^* a^2} \left[ 2 - \cos(k_x a) - \cos(k_y a) \right] \sigma_z. \quad (15)$$

Then, $\Delta E_{\pm}^{\text{latt}} = 4(m^* a^2)^{-1}$ and $8(m^* a^2)^{-1}$ determine the gap sizes of the Dirac fermion doublers at the corners of the Brillouin zone. In fig. 4, we show how the fermion doublers at the boundary of the Brillouin zone become massive for different choices of $\Delta E_{\pm}^{\text{latt}}$, and that the Weyl cone at $k_x = k_y = 0$ is tilted through $u_x$ and $u_y$.

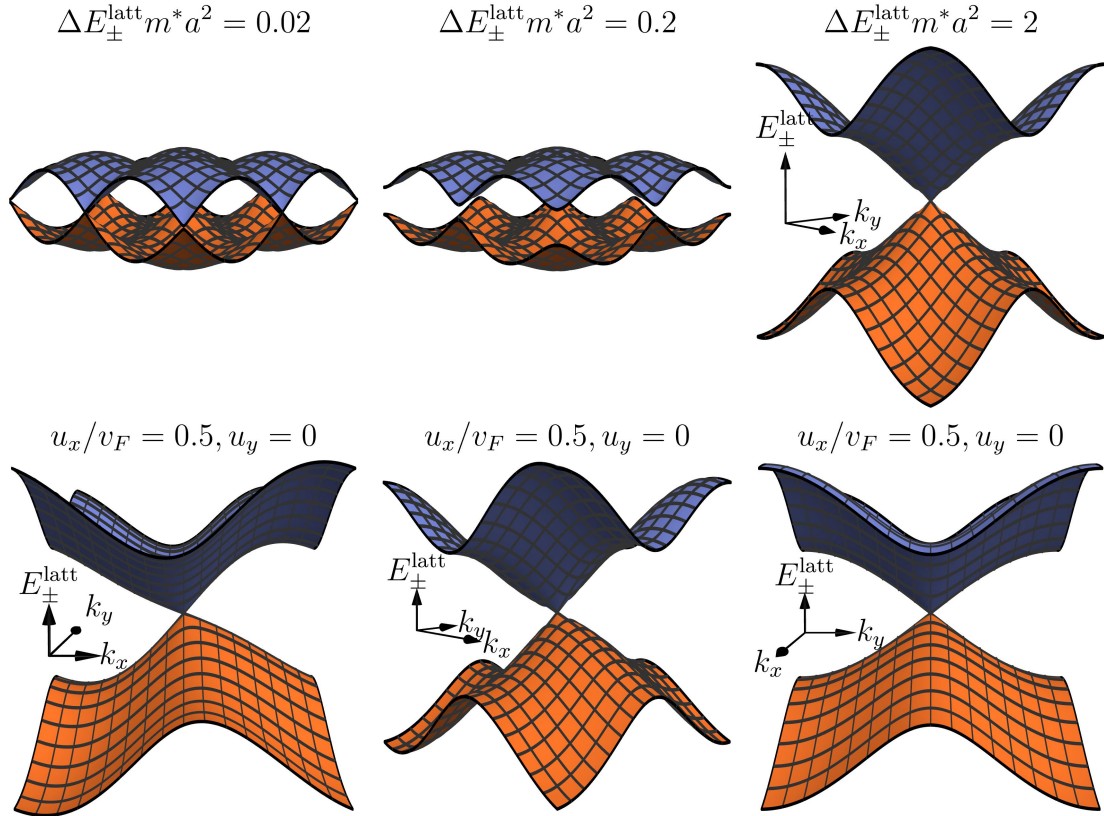

Figure 4: Dispersion relation in the first Brillouin zone of the effective 2D lattice model. In all figures, we fix $k_z = -k_0$, such that the effective 2D lattice model reduces to a single Weyl node at zero energy, which coincides with the low energy continuum model in eq. (1). The cone can then be tilted along $k_x$ and $k_y$ through $u_x$ and $u_y$, respectively. In the top row, $\boldsymbol{u} = \boldsymbol{0}$ and we show the gapping of the Dirac fermion doublers on the corners of the Brillouin zone. In the bottom row, we fix $\Delta E_\pm^{\text{latt}} m^* a^2 = 2$, $u_x = 0.5 v_F$, $u_y = 0$ and rotate the coordinate system.

We continue by transforming the Bloch Hamiltonian in eq. (15) to real space in the $(x, y)$-plane. This results in the following effective 2D lattice Hamiltonian

$$\widehat{H}_{\text{latt}} = \frac{1}{a} \sum_{i \in \{x,y\}} \sum_{\boldsymbol{r}' - \boldsymbol{r} = \hat{\boldsymbol{i}}} \left( \hat{c}_{\boldsymbol{r}}^\dagger \left[ i u_i \sigma_0/2 + i v_F \sigma_i/2 - \frac{1}{2m^* a} \sigma_z \right] \hat{c}_{\boldsymbol{r}'} + h.c. \right) + \frac{2}{m^* a^2} \sum_{\boldsymbol{r}} \hat{c}_{\boldsymbol{r}}^\dagger \sigma_z \hat{c}_{\boldsymbol{r}} \,. \quad (16)$$

We now promote the constant tilt to a slowly-varying tilt profile $u_i \to u_i(\boldsymbol{r} - \boldsymbol{r}_0)$ discretized on the lattice, where $\boldsymbol{r}_0$ denotes the spatial displacement of the tilt center. We note that the class of tilt profiles considered in section 3 preserves $k_z$ as a good quantum number. To avoid placing the singularity at a particular lattice node, we consider the average tilt between two lattice nodes, i.e.,

$$\widehat{H}_{\text{latt}} = \frac{1}{a} \sum_{i \in \{x,y\}} \sum_{\boldsymbol{r}' - \boldsymbol{r} = \hat{\boldsymbol{i}}} \hat{c}_{\boldsymbol{r}}^\dagger \left[ i u_i \left( \frac{\boldsymbol{r} + \boldsymbol{r}'}{2} - \boldsymbol{r}_0 \right) \sigma_0/2 + i v_F \sigma_i/2 - \frac{1}{2m^* a} \sigma_z \right] \hat{c}_{\boldsymbol{r}'} + h.c.$$

$$+ \frac{2}{m^* a^2} \sum_{\boldsymbol{r}} \hat{c}_{\boldsymbol{r}}^\dagger \sigma_z \hat{c}_{\boldsymbol{r}} \,. \quad (17)$$

Finally, we define the real-space Hamiltonian matrix $H_{\boldsymbol{r}, \boldsymbol{r}'}$ implicitly through the identity $\widehat{H}_{\text{latt}} = \sum_{\boldsymbol{r}, \boldsymbol{r}'} \hat{c}_{\boldsymbol{r}}^\dagger H_{\boldsymbol{r}, \boldsymbol{r}'} \hat{c}_{\boldsymbol{r}'}$.

We want to compare wavepacket dynamics in the microscopic lattice model with the results obtained from the semiclassical equations of motion in eq. (3). For the sake of clarity, these equations were previously evaluated using a linearized dispersion in section 3. Linearization of the bands, however, is only valid at low energies. In the lattice model, the curvature of the Weyl cone can become important. Deviations are expected between the outcomes obtained from the linearized and lattice model at finite energies even on the level of semiclassics. Therefore, we first investigate the validity of the predictions derived from the linearized dispersion by comparing trajectories and lensing angles to those obtained from the full lattice dispersion. For $m^* a v_F = 1$, the energy of the lattice model in eq. (17) reads

$$E_\pm^{\text{latt}} = \sum_i \frac{u_i(\rho, \phi)}{a} \sin(k_i a) \pm \varepsilon^{\text{latt}}(\boldsymbol{k}), \tag{18}$$

$$\varepsilon^{\text{latt}}(\boldsymbol{k}) = \frac{v_F}{a} \sqrt{6 - 4\cos(k_y a) + 2\cos(k_x a)(\cos(k_y a) - 2)}, \tag{19}$$

where $u_x = u(\rho)\cos\phi$ and $u_y = u(\rho)\sin\phi$. All trajectories presented in section 4 are obtained by numerically integrating the coupled equations of motion in eq. (3). For simplicity, we here assume a vanishing Berry curvature $\boldsymbol{\Omega} = 0$, because we have checked that it affects the planar motion only negligibly.

The lattice dispersion is in general not axially symmetric around the $z$-axis. This can be further investigated by the Taylor expansion about a general momentum $\boldsymbol{\kappa}$

$$\varepsilon^{\text{latt}}(\boldsymbol{k}) = \boldsymbol{v}_{\text{eff}}(\boldsymbol{\kappa}) \cdot (\boldsymbol{k} - \boldsymbol{\kappa}) + \varepsilon^{\text{latt}}(\boldsymbol{\kappa}) + \mathcal{O}\left((\boldsymbol{k} - \boldsymbol{\kappa})^2\right), \tag{20}$$

where we introduced the effective velocity field

$$\boldsymbol{v}_{\text{eff}}(\boldsymbol{\kappa}) = \frac{v_F^2}{a \varepsilon^{\text{latt}}(\boldsymbol{\kappa})} \left[ \sin(\kappa_x a)(2 - \cos(\kappa_y a))\hat{\boldsymbol{x}} + \sin(\kappa_y a)(2 - \cos(\kappa_x a))\hat{\boldsymbol{y}} \right]. \tag{21}$$

The constant $\varepsilon^{\text{latt}}(\boldsymbol{\kappa})$ is not important for the dynamics and thus neglected. One can confirm that $\boldsymbol{v}_{\text{eff}} \approx v_F \hat{\boldsymbol{\kappa}}$ up to third order around $\boldsymbol{\kappa} = 0$, and therefore $\boldsymbol{v}_{\text{eff}} = v_F \hat{\boldsymbol{k}}$ in the vicinity of $\varepsilon^{\text{latt}} = 0$. The anisotropy of the dispersion and velocity field increases towards the boundary of the Brillouin zone. For the linear dispersion, the solutions of $\boldsymbol{v}_g(\boldsymbol{r}_t, \boldsymbol{k}) = \boldsymbol{u}(\boldsymbol{r}_t) + v_F \hat{\boldsymbol{k}} = 0$ are clearly momentum independent for a cylindrically symmetric tilt $\boldsymbol{u}(\boldsymbol{r}) = u(\rho)\hat{\boldsymbol{\rho}}$ and $u(\rho) = -v_F(\rho/\rho_t)^\alpha$, giving rise to a horizon: particles can only enter and never leave the regime $\rho < \rho_t$. Instead, for the lattice model, the condition $\boldsymbol{v}_g(\boldsymbol{r}_t, \boldsymbol{k}) = \boldsymbol{u}(\boldsymbol{r}_t) + \boldsymbol{v}_{\text{eff}}(\boldsymbol{k}) = 0$ has momentum-dependent solutions $\boldsymbol{r}_t(\boldsymbol{k})$. The horizon associated with a vanishing group velocity is thus washed out to constraints on positions that are not allowed to be crossed by specific trajectories with momentum $\boldsymbol{k}$. For axial tilt profiles, only $k_z$ is a conserved quantity, and the constraints are thus dynamic. Although the horizon "melts" due to the curvature at higher energies, the associated length scales $\rho_t$ and $\rho_s$ still mark the mesoscopic length scale where equation of motion semiclassics breaks down. This is most prominently visible in section 4.3, where we compare semiclassical trajectories and microscopic simulations.

The isotropic linear approximation remains valid as long as $|\boldsymbol{v}_{\text{eff}}(\boldsymbol{k}(t)) - v_F \hat{\boldsymbol{k}}(t)| \ll v_F$. Since the separatrix of the linearized model is an unstable orbit, it cannot be recovered by the trajectories from the lattice dispersion. For strong lensing the linearized approximation can only hold if the evolution of $\boldsymbol{k}(t)$ resides in the isotropic region of $\boldsymbol{v}_{\text{eff}}$ at very low energies $\kappa \approx 0$.

The lattice dispersion has especially noticeable consequences for trajectories crossing the separatrix, as is shown in panel ($a$) of fig. 5. At small energies, $Em^* a^2 \lesssim 0.01$, the trajectories obtained using the linearized dispersion match extremely well with the ones found using the full lattice dispersion. Note that we terminate the semiclassical trajectory for $Em^* a^2 = 0.01$ at

Figure 5: In panel $(a)$ and $(b)$ we display semiclassical trajectories obtained by the linearized (dashed line) and lattice (solid lines) dispersion relation for $\alpha = 1/2$ at different energies. Initial coordinates are chosen at $x = 0$, and initial momenta are $k_y = 0$, with $k_x > 0$ matching the energies indicated in the legend. For the linearized trajectory, we fixed $k_y = 0$ and $k_x \rho_t = 2$. Panel $(a)$ shows that capture at low energies turns into slingshot trajectories when we use the lattice dispersion. For even larger energies, the trajectories continuously become weakly lensed ones. Panel $(b)$ displays the energy-dependence of lensed trajectories, which shows only minute deviations. In $(c)$, we show the lensing angle for initial conditions $k_x = -k_y$, and for $(d)$ for $k_y = 0$. The universal asymptotic decay of eq. (12) is recovered for all energies.

the center of the tilt, as the equations of motion suffer from numerical instabilities (stiffness) in the vicinity of the divergence. Since the total energy must be conserved, we introduce a threshold on the energy error of the simulated trajectories which detects these instabilities. For all simulations, we accept times which satisfy $|(E_\pm^{\text{latt}}(t) - E_\pm^{\text{latt}}(0))| < 10^{-5} |E_\pm^{\text{latt}}(0)|$.

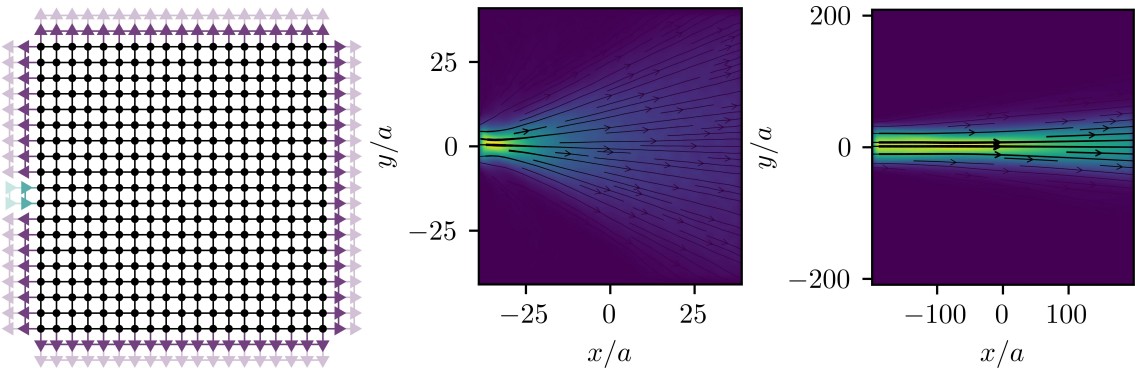

Figure 6: The left panel depicts the microscopic setup described in the main text for $L_x = L_y = 19$. The lattice sites of the center region are represented by black dots, and hopping events are depicted by black lines. The purple and green triangles and lines represent sites and hopping events of the semi-infinite leads, which are coupled to the edge sites of the center region. We inject particles from the blue lead, which may leak out in the purple lead regions. The remaining panels display the charge density in color (arbitrary units) for a Weyl semimetal without tilt, superimposed with streamlines corresponding to the charge current vector field of a scattering state with energy $Em^*a^2 = 0.8$ and initial momentum $k_x a = 0.832$ for different system dimensions. The injected ray of particles remains focused within the scattering region when the system lengths exceed 400 sites.

Finally, panel $(b)$ of fig. 5 shows lensing trajectories obtained using the lattice dispersion and the same energies as in panel $(a)$. We find that the lattice dispersion only affects these trajectories quantitatively. For trajectories traversing far away from the tilt center, the linear isotropic dispersion is thus always a good approximation. We show this explicitly in fig. 5 panels $(b)$-$(d)$, for which we present trajectories and deflection angles across all relevant energy scales of the lattice model. The numerical solutions we obtained from the equations of motion of the lattice model yield the same universal asymptotic scaling $\varphi \propto (u(\rho_p)/v_F)^2$ we found in case of the linearized dispersion at $k_z = 0$ (see eq. (12)).

In conclusion, we find that the semiclassical equations of motion using the lattice dispersion deform capture into slingshot trajectories at energies away from $E = 0$, while preserving the asymptotic behavior for large $\rho_p \gg \rho_t$. We note that under special circumstances (when the wavepacket is injected straight into the tilt center, see section 4.3), numerical instabilities can still occur at high energies. At energies $Em^*a^2 \sim 0.1$ and boundary conditions that are not directed straight into the tilt center, we do not find capture trajectories, or numerical instabilities. However, we still note the presence of slingshot trajectories. Here, we define a slingshot trajectory as having a lensing angle $\varphi > \pi/2$. The presence of slingshot trajectories at high energies serves as a fingerprint for a separatrix and captured trajectories at low energies. For the comparison between semiclassical trajectories with the microscopic simulations, we thus use the full lattice dispersion in eq. (3) instead of the linearized dispersion.

## 4.2 The microscopic setup

For a fully microscopic analysis, we use the KWANT library [53] to model a rectangular setup consisting of $(L_x, L_y)$ lattice sites, coupled to six semi-infinite leads. Within the leads, we assume a vanishing tilt profile. We choose one semi-infinite lead along the $-\hat{x}$ direction narrow compared to the diameter of the scattering region ($\approx 0.1L_y$), which allows us to inject wavepackets with a well-defined average initial position and spread along the $\hat{y}$-direction. The

other leads along $\pm\hat{y}$ and $\pm\hat{x}$ fully cover the remaining edge sites, such that injected waves may move freely and escape the central region.

For a given system size and power-law decay $\alpha$, unwanted interference effects caused by reflections from the boundary are suppressed if the radius of the tilt profile satisfies $\rho_t \ll L_m a$, where $L_m a$ is the minimum distance between the tilt center and the boundary site where the injected wave leaks outside the central region. Each one of the semi-infinite leads has $2N_l$ bands, where $N_l$ is the number of sites connecting lead $l$ with the central scattering region. Up to finite-size effects, the bands are the transverse projections of the dispersion presented in fig. 4. Since we set $\boldsymbol{u} = \boldsymbol{0}$ in the leads, the resulting dispersion relations of all leads are equivalent upon exchanging $k_x \leftrightarrow k_y$ appropriately. Note that the chosen device setup does not feature topological bound modes since all edge sites are connected to a lead. A small device setup for $L_x = L_y = 19$ is depicted schematically in fig. 6 (left). For small systems, the injected wave is spread across the full scattering region after reaching the center of the scattering region, which is visible in the charge current density, plotted in fig. 6 (center). If the system length exceeds 400 sites, an injected wave with energy $Em^*a^2 = 0.8$ stays focussed across the center region, visible in fig. 6 (right). To approach the semiclassical limit, the length scales set by the tilt must be much smaller than the lattice spacing $\rho_t \gg a$ in the microscopic setup.

Observables such as the charge density and current are computed from the vector consisting of the wavefunction components of a scattering state $|\psi\rangle$, i.e., $\boldsymbol{\psi_r} = \langle \boldsymbol{r}|\psi\rangle$. We define the more general observables

$$n_{i,\boldsymbol{r}} = \boldsymbol{\psi}_{\boldsymbol{r}}^{\dagger}\sigma_i\boldsymbol{\psi}_{\boldsymbol{r}}, \quad J_{i,\boldsymbol{r},\boldsymbol{r}'} = \mathrm{i}\left(\boldsymbol{\psi}_{\boldsymbol{r}'}^{\dagger}(H_{\boldsymbol{r},\boldsymbol{r}'})^{\dagger}\sigma_i\boldsymbol{\psi}_{\boldsymbol{r}} - \boldsymbol{\psi}_{\boldsymbol{r}}^{\dagger}\sigma_i H_{\boldsymbol{r},\boldsymbol{r}'}\boldsymbol{\psi}_{\boldsymbol{r}'}\right), \tag{22}$$

corresponding the local density and current with respect to the Pauli matrix $\sigma_i$, respectively. Because of the finite system size, the wavepacket will never reach its asymptotic deflection angle. To compensate this limitation, we compare the local charge density with the semiclassical trajectories for different initial conditions, rather than deflection angles. This will be discussed in the next section.

## 4.3 Numerical results

In this section, we study numerically the behavior of injected waves upon fixing $\rho_t$ and reducing the distance between the injected wavepacket and the tilt center. The analysis is based on the computation of the local charge density for different scattering states, which are compared to numerical solutions of the semiclassical equations of motion [using the lattice dispersion relation, eq. (19)] with equivalent initial conditions. In particular, the initial momenta for the semiclassical equations of motions are taken from the injecting lead modes, and for the initial positions we take the edge sites of the injecting lead. To avoid overcrowded figures, we overlay only seven semiclassical trajectories on top of the charge density. Numerical instabilities in the solution of the semiclassical equations can also be caught by limiting the trajectory velocity, $v_{\max} < 100\,v_F$. The tilt velocity at $u(\rho_t) = -v_F$ is fixed to the group velocity at energy $E = 0$. We acknowledge that this imposes a slight inconsistency: in the lattice computations (semiclassical and microscopic), the slope of the Weyl cone is not constant and gains a momentum dependence at higher energies. Consequently, states at different momenta experience a different type II region in general. However, a detailed investigation of the trajectory-dependent tilt length scales is beyond the scope of this section, and we continue by using static reference length scales derived from the low-energy theory with constant slope. Furthermore, the width of the injected wave is on the same order of magnitude as the length scale of the tilt profile. We thus expect scattering events for short distances between the average position of the injected wave and the center of the tilt profile.

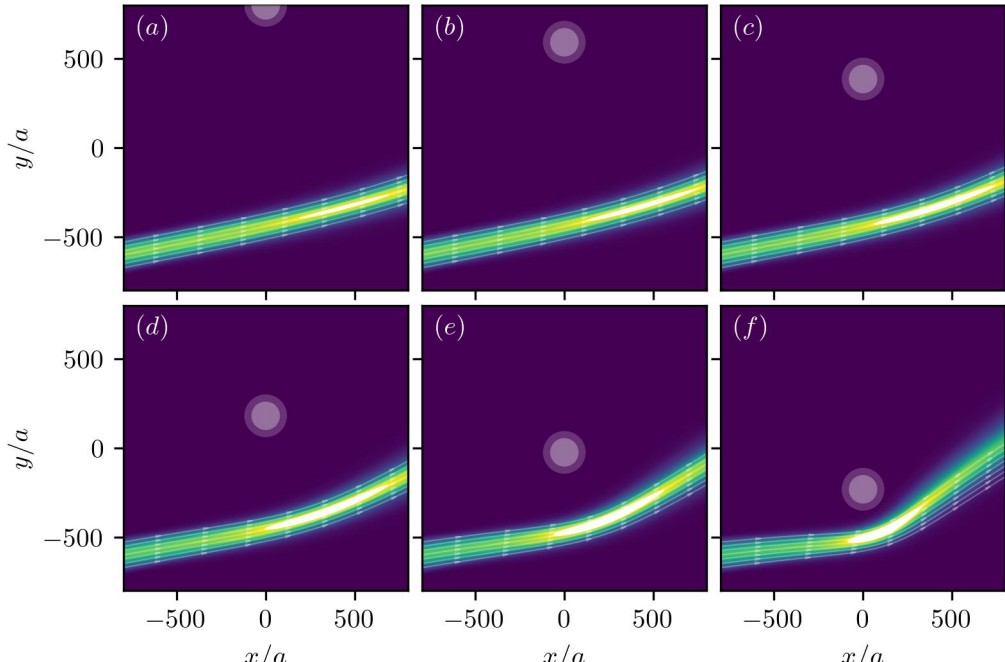

Figure 7: Lensing scenarios for different initial conditions. The length scales set by the tilt, $\rho_t$ and $\rho_s = 3\rho_t/2$, are marked by purple disks. The energy is fixed to $Em^*a^2 = 0.8$, such that the initial momenta are $k_y = 0$ and $k_x a = 0.823$. We find excellent agreement between semiclassical trajectories and microscopic simulations for large distances between tilt center and injected wave. We find quantitative deviations between semiclassical trajectories and microscopic simulations if the distance between wave packet and tilt center is on the order of $\rho_s$.

Figure 7 displays the local charge density of a scattering state, subject to a tilt profile with $\alpha = 1/2$ at a high energy ($Em^*a^2 = 0.8$). The tilt length scales $\rho_t$ (event horizon analogue) and $\rho_s = 3\rho_t/2$ (separatrix for $\alpha = 1/2$) are marked by white semi-transparent disks. Here and in the following, we omit to superimpose the current vector field, because the streamlines would be largely overlapping with the semiclassical trajectories outlined as white arrows. We find excellent agreement when the distance between the center of the wavepacket and the tilt center is much larger than the separatrix radius $\rho_s$. In the vicinity of the separatrix, we find quantitative deviations between semiclassical trajectories and the trajectory of the microscopic simulation. For even smaller distances (not shown in fig. 7), the wavepacket is heavily scattered and splits into jets. To investigate these scattering events in greater detail, we consider a tilt profile that decays faster, which allows us to separate the length scales between tilt and width of the wavepacket.

In fig. 8, we display various simulations for a tilt profile with $\alpha = 1$, at a lower energy $Em^*a^2 = 0.4$. Overall, we find qualitatively the same physical trajectories for all $\alpha$, but a fast tilt decay allows us to increase the radius $\rho_t$ significantly without the introduction of unwanted boundary reflections due to the non-vanishing tilt at the system-lead interface. For large distances between wavepacket and tilt center, $\rho \gg \rho_t$, we again observe that the injected waves establish smooth deflections (panels $(a)$, $(b)$ and $(c)$). The deflection observed in the microscopic simulation is in excellent agreement with the semiclassical predictions. For smaller distances, $\rho \sim \rho_t$, shown in panels $(d)$ to $(i)$, the injected wave separates into jets, which traverse very similar to the predictions from semiclassical trajectories. At locations where different jets are crossing, we observe additional interference patterns. We further observe clear slingshot trajectories in panels $(f)$, $(g)$ and $(h)$, which signal the presence of a separatrix at low

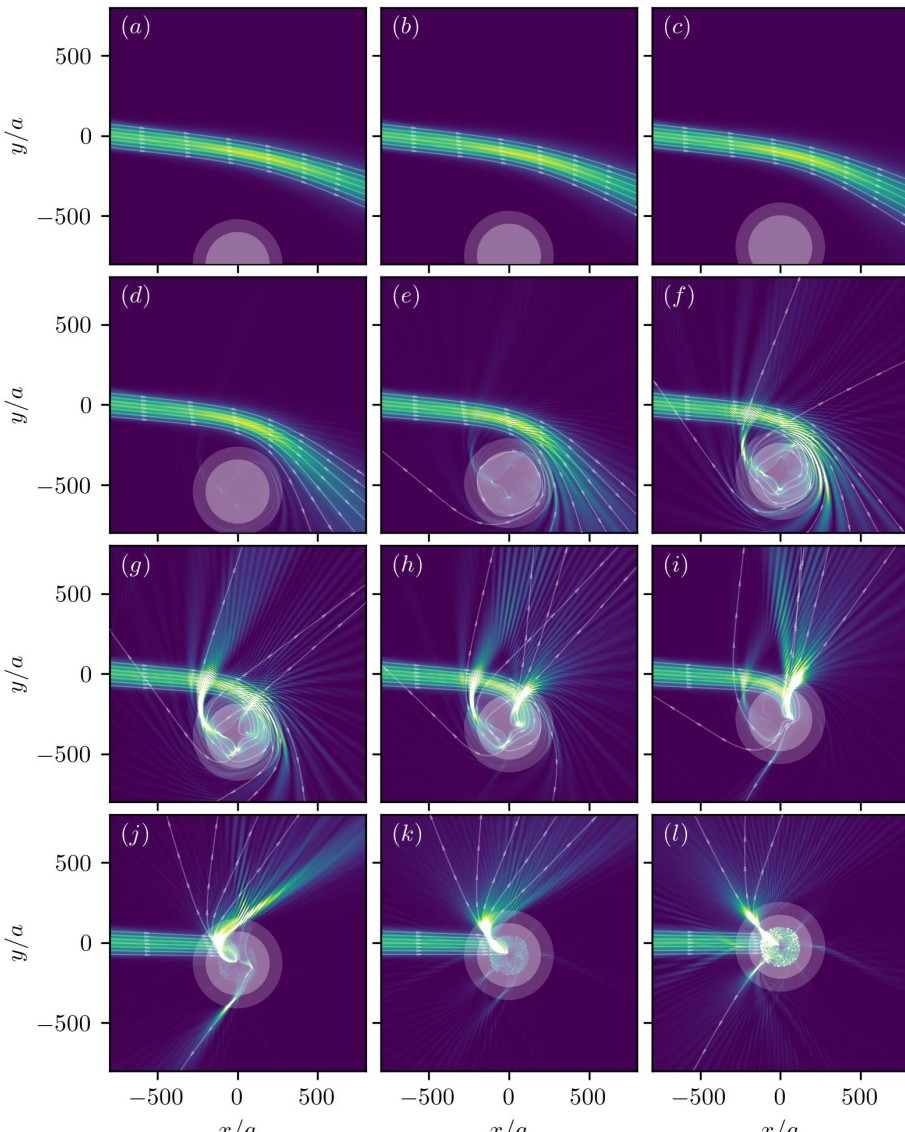

Figure 8: Local charge density of scattering states for different initial conditions with $\alpha = 1$. The length scales set by the tilt, $\rho_t$ and $\rho_s = \sqrt{2}\rho_t$, are marked by purple disks. The energy is fixed to $Em^*a^2 = 0.4$, such that the initial momenta are $k_y = 0$, $k_x a = 0.403$. We find excellent agreement between semiclassical trajectories (white semi-transparent lines) obtained by numerical solutions of the equations of motion, and the microscopic trajectory (color gradient) mapped out by the local charge density of the scattering states.

energies. In panels $(k)$ and $(l)$, we display results of a third regime where the distance between injected wave and tilt center vanishes. In panel $(l)$, we note that only 4 of the 7 semiclassical trajectories escape the overtilted regime. The remaining 3 trajectories are suffering from the aforementioned numerical instabilities and the evaluation is stopped in the vicinity of the tilt center (where the velocity exceeds $100\, v_F$). Although the dominant jets of the injected wave that escape the tilt center clearly follow semiclassical trajectories, we additionally observe a circular domain of size $\approx 0.8\rho_t$ surrounding the tilt center where the injected wave is heavily scattered. Although the size and shape of the scattering domain suggests that it relates to the radius of an effective analog horizon at high energy scales, we stress that any such statement is a conjecture at this point, and remains to be investigated in future works.

In conclusion, our microscopic lattice simulations show that trajectories satisfying the semi-classical equations of motion provide an excellent description of the motion of wavepackets sufficiently far from the strongly tilted region. This means that the physically transparent picture of motion in a gravitational field can be used to engineer electronic lenses realizing the solid state analogue of gravitational lensing. At high energies, the low energy description breaks down partially, and quantitative agreement is only recovered if we account for the lattice dispersion in eq. (3). At the same time, our simulations show that this analogy breaks down when the wavepacket comes too close to the strongly overtilted region mimicking the black hole (the critical distance here is of the order of the separatrix discussed in section 3). On these lengths scales, lattice effects break the analogy to black hole physics. Put in simple terms, the effective black hole fades away if it is probed closely, much like a mirage.

# 5  Motion along the cylinder axis: Berry curvature, spin dynamics and transverse shift

In section 3, we analyzed semi-classical equations of motion in Weyl semimetals with tilt profiles having axial symmetry, and focused on the dynamics in the $(\rho, \phi)$-plane perpendicular to the cylinder axis. This was motivated by the observation that for initial momentum $k_z = 0$ along the $z$-direction, $\dot{z}$ decouples from the other coordinates. We want to conclude by also analyzing the motion along $z$ based in the semiclassical equations of motion for a linearized spectrum (keeping $k_z = 0$, which implies initial $\dot{z} = 0$ asymptotically far from the strongly tilted region).

From eq. (4), we find

$$\dot{z} = -\frac{\chi}{2k^3}\left(\frac{\partial u}{\partial \rho} - \frac{u}{\rho}\right)(\boldsymbol{k} \cdot \hat{\boldsymbol{\rho}})\frac{L_z}{\rho}. \tag{23}$$

This equation shows that for $k_z = 0$, the motion along $z$ is exclusively due to the anomalous velocity originating from the finite Berry curvature of Weyl semimetals, i.e., the term $\dot{\boldsymbol{k}} \times \boldsymbol{\Omega}$ in eq. (4) (this is the only term that explicitly depends on the chirality $\chi$). Depending on the origin of $\dot{\boldsymbol{k}}$, this term has been connected to a chirality-dependent Hall effect [54, 55], a Hall effect induced by strain and an applied electric field [56] and the topological Imbert-Federov effect [57]. Transverse shifts have also be found in impurity scattering [58]. We note that other Berry curvature components, e.g. $\boldsymbol{\Omega}_{rk}$ and $\boldsymbol{\Omega}_{rr}$, could also lead to transverse shifts when the Fermi velocity changes [54, 55]. The Hall effect and transverse response has also been reported in tilted Weyl semimetals with applied electromagnetic fields [59, 60] and for tunneling across potential barriers [61]. In the gravitational context, the transverse shift of photons travelling in a curved spacetime is known as the gravitational spin Hall effect [62–64]. Our analysis extends the discussion of transverse shift to the case of inhomogeneously tilted Weyl nodes.

To connect with section 3, we stress that eq. (23) shows the radial position of the separatrix not to be affected by Berry curvature effects. The motion of a wavepacket on the separatrix, however, is strongly affected: depending on the chirality, the wavepacket will either spiral upwards or downwards along the $z$ direction while staying at a fixed radius (this is shown explicitly in fig. 10).

For the case of lensing, a wavepacket approaching from and escaping to infinity undergoes a finite shift along the $z$-axis during its interaction with the tilt profile as shown in fig. 9. The asymptotic value of the transverse shift can be calculated from $\Delta z = \int d\rho\, \dot{z}/\dot{\rho}$. An explicit evaluation for lensed trajectories that stay sufficiently far away from the tilt center such that

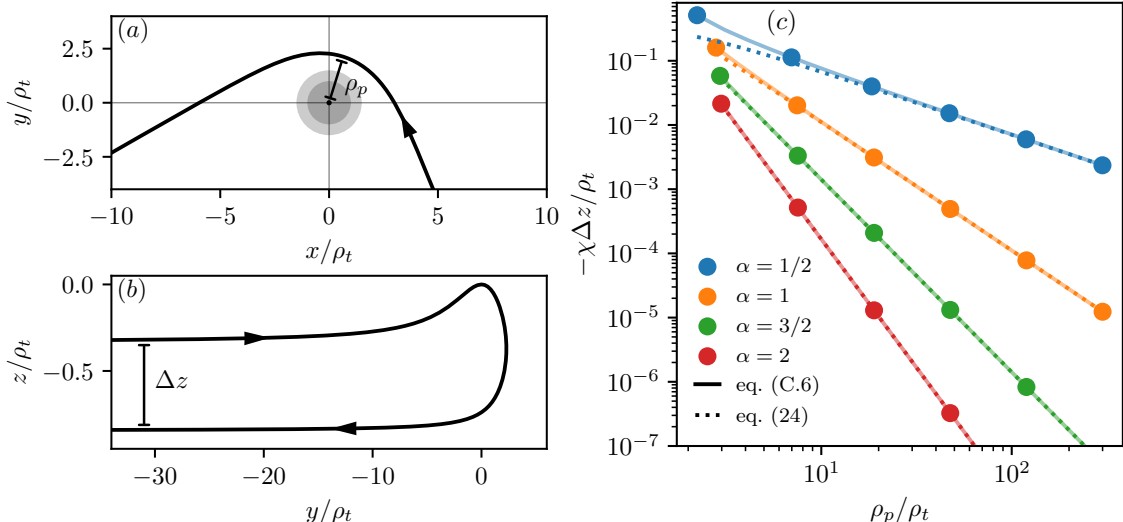

Figure 9: Transverse shift for $\chi = +1$. In panels $(a)$ and $(b)$, we show the $x, y$ and $y, z$ projections of an example trajectory, corresponding to $\alpha = 1/2$, $\rho_p/\rho_t = 2.23$ and $\Delta z = -0.512$. In $(c)$, we show the transverse shifts $-\Delta z$ for chirality $\chi = +1$, and compare results obtained by numerically integrating the semiclassical equations of motion (dots) with numerical integration of eq. (C.6) (solid lines) and the approximation in eq. (24) (dotted line) for different $\alpha$ (different colors).

$\rho_p/\rho_t \gg 1$ is given in appendix C. We find

$$\Delta z \approx -\frac{\chi}{L_z} \rho_p \frac{u(\rho_p)^2}{v_F^2} \frac{\sqrt{\pi}\,\Gamma\left(\alpha + \frac{1}{2}\right)}{2\,\Gamma(\alpha)}. \tag{24}$$

The transverse motion due to Berry curvature effects also manifests in transport as non-equilibrium "anomalous currents" when the system is coupled to external magnetic fields and mechanical rotation. These lead to the chiral magnetic and the chiral vortical effects in Weyl systems and have been extensively studied in the literature [65–72].

## 5.1 Transverse shift, spin precession, and local Lorentz boosts

As mentioned in the last paragraphs, the motion along $z$ results from the anomalous velocity $\dot{\boldsymbol{k}} \times \boldsymbol{\Omega}(\boldsymbol{k})$ in eq. (3). The physics of the anomalous velocity is in fact intricately related to the topological spin-transport of massless particles [73–77]. To appreciate this connection, we recall that the semiclassical equations of motion we analyze here are the ones of a massless spinning particle, i.e a Weyl fermion [78, 79].

An elementary calculation shows that the spin evolves as

$$\frac{d\boldsymbol{s}}{dt} = \frac{\chi}{2} \frac{(\boldsymbol{k} \times \dot{\boldsymbol{k}})}{k^3} \times \boldsymbol{k} = \boldsymbol{\omega}_{sp} \times \boldsymbol{s}, \tag{25}$$

where we have introduced a frequency $\boldsymbol{\omega}_{sp} = (\boldsymbol{k} \times \dot{\boldsymbol{k}})/k^2$. The dynamics of $\boldsymbol{s}$ thus takes the form of a precession. Specifically for the case of an axially symmetric tilt, one finds (for $k_z = 0$)

$$\boldsymbol{\omega}_{sp} = \frac{(\boldsymbol{k} \cdot \hat{\boldsymbol{\rho}})}{\rho\, k^2} \left(\frac{\partial u}{\partial \rho} - \frac{u}{\rho}\right) L_z \hat{\boldsymbol{z}}. \tag{26}$$

Using the explicit form of the Berry curvature allows one to directly connect the anomalous velocity $\dot{\boldsymbol{r}}_A$ to this precession as

$$\dot{\boldsymbol{r}}_A = \dot{\boldsymbol{k}} \times \boldsymbol{\Omega} = -\frac{\chi}{2\,k} \boldsymbol{\omega}_{sp}. \tag{27}$$

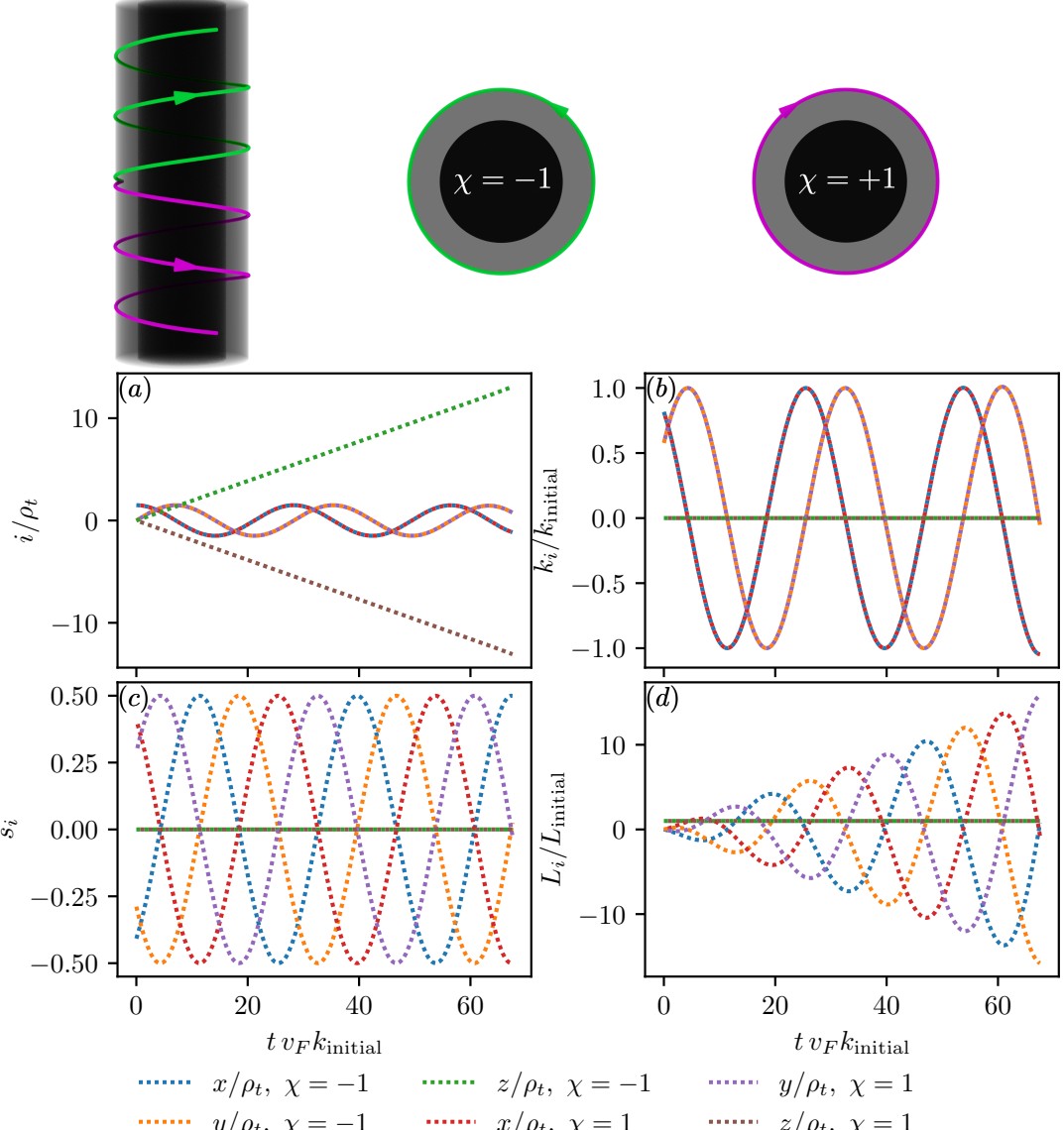

Figure 10: Trajectories for different chiralities with initial coordinates chosen from the separatrix. We take $\alpha = 1/2$, and choose $(x, y, z)/\rho_t = (3/2, 0, 0)$, $(k_x, k_y, k_z)\rho_t = (\sqrt{2}/3, 1/3, 0)$. The black and gray cylinder represent the analog horizon and photon cylinder. The purple and green trajectory corresponds to $\chi = +1, -1$, respectively, and orthographic projections from $\pm \hat{z}$ are shown in the top row. In panels $(a)$ and $(b)$, we show that planar components of the coordinates are identical for the two chiralities, and oscillate around the $\hat{z}$ axis. The transverse motion, the planar spins, and the planar angular momenta are sensitive to the chirality of the Weyl node (panels $(c)$ and $(d)$). The oscillation of planar moments results in a precession of the spin around the $\hat{z}$ axis (panel $(c)$). The planar angular moments also show an oscillatory behavior, but increase in magnitude (panel $(d)$).

Thus the transverse shift from the anomalous velocity is intricately related to the spin precession. The spin dynamics induced through the tilt profile contributes to the evolution of the orbital angular momentum through the anomalous velocity term. The dynamics of position, momentum, spin and orbital angular momentum of both chiralities for a tilt profile with axial symmetry are shown in fig. 10 for a trajectory from the separatrix and in fig. 11 for a lensing

scenario, both with $k_z = 0$. The $s_z$, $L_z$ components are conserved and the spin and angular moment dynamics is restricted to the planar components $s_x$, $s_y$ and $L_x$, $L_y$ in both cases. For a trajectory chosen from the separatrix, shown in fig. 10 (panel $(a)$), the two chiralities wind in opposite directions along the $z$-axis. The spin precession, caused by the planar precession of $\boldsymbol{k}$, is explicitly shown in panels $(b)$ and $(c)$. The dynamics of orbital angular momenta is shown in panel $(d)$. Since the orbital angular momentum is not conserved along the planar directions, the absolute magnitude of $\boldsymbol{L}$ increases with time. The dynamics of a lensing trajectory (see fig. 11) indicates that there is a slim window, confined around the periapsis, where non-conserved components of momentum, spin, and angular momentum undergo a rapid change. The transverse shift along $z$-direction asymptotes to the value calculated in eq. (24) and presented in fig. 9. A notable feature in fig. 11 panels $(a)$, $(c)$ and $(d)$ are the crossings of $z$, $s_x$, $s_y$, $L_x$ and $L_y$ for trajectories of opposite chiralities. As can be seen in panels $(c)$ and $(d)$, an incoming trajectory with spin along one of the planar components undergoes an overall flip and obtains a higher orbital angular momentum as it escapes out to infinity.

The connection between the anomalous velocity and spin precession has long been discussed in the context of relativistic quantum mechanics. For Weyl fermions, relativistic quantum-mechanical particles with spin, the fundamental dynamical quantities have precisely been identified as being momentum and spin [80,81]. When deriving semiclassical equations of motion for position and momentum as seen from a lab-frame, an anomalous velocity term related to the spin evolution is obtained [74]. More generally, Lorentz covariance of the relativistic Weyl equation, its inheritance to the Weyl Hamiltonian and the ensuing consequences in the semiclassical dynamics have been topics of fundamental importance [66,73–75,78,79,82,83]. They have been extensively studied in their relation to topological spin transport [77,84], the spin Hall effect of fermions [85], photons [63,64,86] and phonons [87], the Imbert-Federov effect, and transverse shifts [88].

An alternative viewpoint on the connection between the anomalous velocity and spin physics is offered by local Lorentz boosts. A key ingredient in the spin dynamics is a nonzero $\dot{\boldsymbol{k}}$, see eq. (25). In our case, $\dot{\boldsymbol{k}}$ is generated by the spatial variation of the tilt profile. The key observation is now that a Weyl Hamiltonian with a tilt $\boldsymbol{u}$ is equivalent to applying Lorentz boosts with velocity $\boldsymbol{u}$ on a usual untilted Weyl system. This fact has already been used in previous studies of transport phenomena in tilted WSMs [89–91]. A particle moving in a spatially varying tilt profile can therefore be understood as undergoing a local Lorentz boost with $\boldsymbol{u}(\boldsymbol{r})$ at every point $\boldsymbol{r}$ as it moves through the tilt profile. Such a continuous sequence of non-collinear Lorentz boosts results in a rotation which underlies the Thomas precession of an electron, and its relation to Berry curvature in semiclassical dynamics has been noted earlier [74].

## 6 Summary and Conclusions

In summary, we have studied electron trajectories in Weyl semimetals with a spatially varying tilt profile. We have used and compared three different approaches to calculating such trajectories for a cylindrically symmetric tilt profile: (i) in the limit where the spectrum near the Weyl node can be approximated as linear, the semiclassical equations of motion can be mapped onto geodesic equations for relativistic particles moving in an effective curved spacetime. Going beyond this linear approximation, we have investigated (ii) a high-energy regularization of semiclassical trajectories using a particular lattice model. Finally, (iii) we have compared these semiclassical trajectories to fully quantum-mechanical scattering states which we calculated numerically for a microscopic lattice model.

For an axially symmetric tilt profile featuring an effective event horizon, the geodesic equations predict the existence of an analog of a "photon sphere" of a black hole, i.e., an unstable

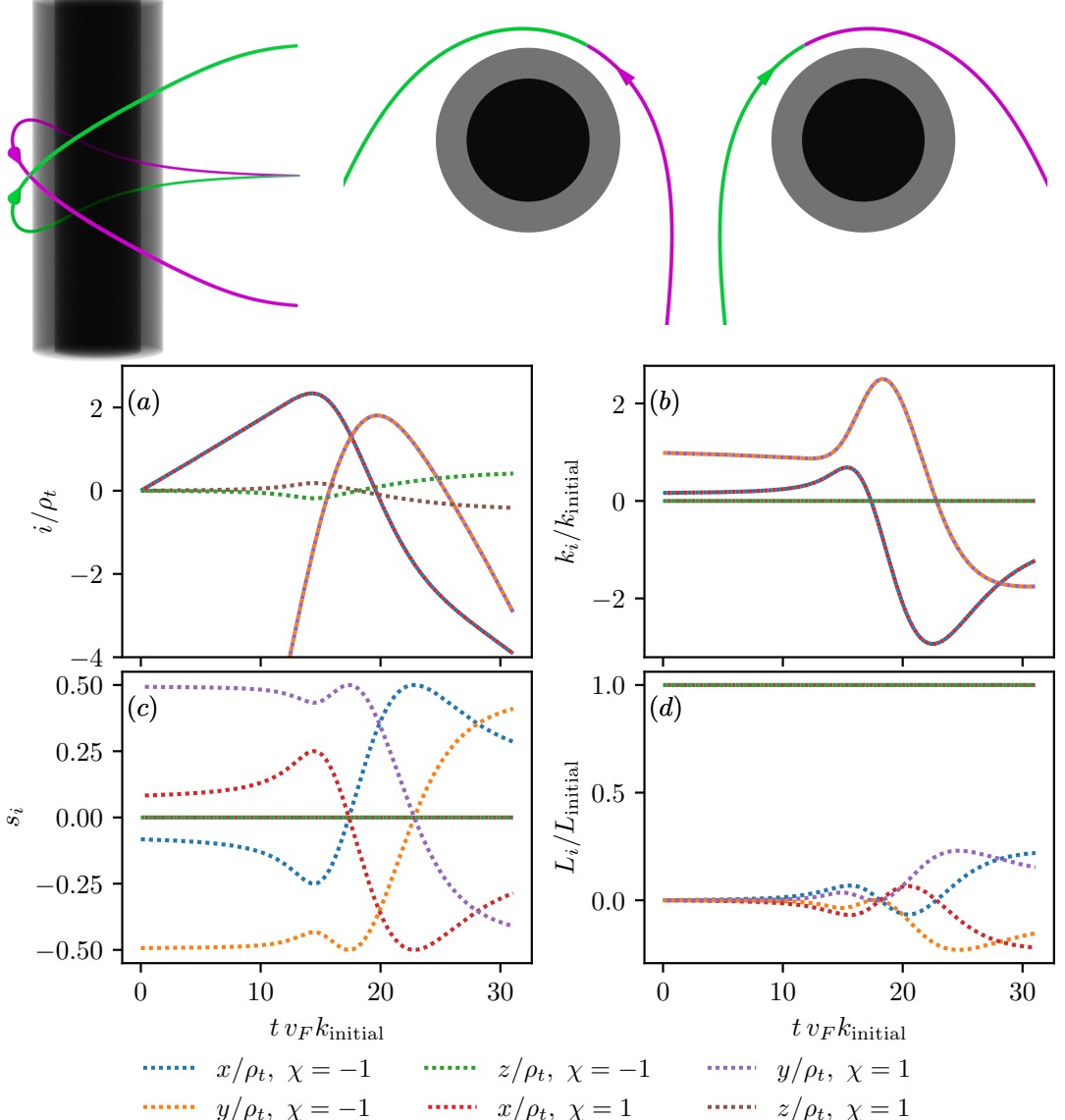

Figure 11: Lensing trajectories for different chiralities. We take $\alpha = 1/2$, and choose $(x, y, z)/\rho_t = (0, -20, 0)$ and $(k_x, k_y, k_z)\rho_t = (0.167, 1, 0)$. The black and gray cylinder represent the analog horizon and photon cylinder. The purple and green trajectory corresponds to $\chi = +1, -1$, respectively, and orthographic projections from $\pm\hat{z}$ are shown in the top row. In panels $(a)$ and $(b)$, we see that the planar components of position and momenta are identical for the two chiralities. The signs of the transverse motion along $\hat{z}$, the planar components of spin, and the orbital angular momentum are directly connected to the chirality of the Weyl node. The planar spin components of a trajectory approaching the origin are flipped in the vicinity of the periapsis, and the orbital angular momentum is increased (panels $(c)$ and $(d)$).

orbit which separates captured trajectories from lensed trajectories, which we term the "photon cylinder". We have confirmed that the solution of the geodesic equations provides an excellent match with the quantum-mechanical solutions as long as the particles stay far away from this photon cylinder. In particular, we showed that the deflection angle computed from trajectories based on the lattice dispersion feature the same universal asymptotic scaling of the deflection angle across all relevant energy scales. Upon closer approach to the photon cylinder,

geodesics start to become inaccurate, but semiclassical trajectories still provide an excellent approximation provided that they are calculated using the lattice dispersion instead of the linearized spectrum. The fact that the tilt profile acts like an analog black hole on electron trajectories far away from it, but that this analog black hole disappears upon closer approach has led us to dub this phenomenon a "black-hole mirage".

Our results extend beyond black-hole analogies in semimetals, since we consider more general tilt profiles which do not reflect Schwarzschild-like metrics. For the linearized dispersion, the general tilt profile gives rise to a horizon, which is "melted" by curvature effects in the lattice model. One of our main conclusions is that the semiclassical approximation based on the linear dispersion breaks down well before the particles reach the mesoscopic length scale $\rho_t$ set by the tilt profile, while the semiclassical approximation based on the lattice dispersion still remains accurate.

Finally, we have studied the effect of the Berry curvature which is necessarily present in systems with Weyl nodes. Focusing on cylindrically symmetric tilt profiles and wavepackets with incident velocity normal to the symmetry axis ($k_z = 0$), we have shown that the in-plane projection of the trajectories is unaffected by the Berry curvature term. In general, the presence of Berry curvature leads to an out-of-plane deflection of the trajectories which is proportional to the chirality of the Weyl cone. Most of the results we presented considered systems with only one Weyl node at the Fermi level. Such band structures can be realized in materials with broken inversion and time-reversal symmetry, in which case a pair of Weyl nodes can be located at different momenta and different energies. However, even in the case of two Weyl nodes at the same energy, i.e., when inversion symmetry is intact, our results remain essentially unchanged. Since the in-plane motion is unaffected by the Berry curvature for $k_z = 0$, it remains identical to the case of a single Weyl cone. If the two nodes are tilted with the same tilt profile $u(\rho)$, as considered in the main text, the out-of-plane motion will be opposite for electrons of opposite chiralities. This means that a cylindrically symmetric tilt profile can act as a chirality-based electron beam splitter. If the nodes are instead tilted in opposite directions, $u(\rho) \to \chi u(\rho)$, eq. (11) and eq. (23) show that the out-of-plane motion will be identical for electrons of opposite chiralities, while the lensing angle remains unchanged.

Spatially-varying tilt profiles can be experimentally realized using, e.g., a magnetic field texture or locally applied strain. For such devices it would be necessary to investigate to which extent impurity scattering affects the bulk transport in these systems. We showed that electron trajectories can be controllably deflected in such tilt profile, potentially enabling "tiltronics" applications.

**Note added** – While preparing this manuscript we became aware of a related work [92] that considers an alternative approach to gravitational lensing in a similar system. As far as our analyses overlap, they reach similar conclusions.

# Acknowledgements

**Funding information**    All authors acknowledge funding from the Luxembourg National Research Fund (FNR) and the Deutsche Forschungsgemeinschaft (DFG) through the CORE grant "Topology in relativistic semimetals (TOPREL)" (FNR project No. C20/MS/14764976 and DFG Project No. 452557895). S.H, C.X. and T.M. furthermore acknowledge support from the DFG through the Emmy Noether Programme ME4844/1, the Collaborative Research Center SFB 1143 (Project No. A04), and the Würzburg-Dresden Cluster of Excellence on Complexity and Topology in Quantum Matter–$ct.qmat$ (EXC 2147, Project No. 390858490). CDB also received funding by the FNR through an INTER mobility grant (FNR project No. INTER/MOBILITY/2021/MS/16515716/ESRMOIRE). For the purpose of open access, the au-

# A Relationship between the geodesic equation and semiclassical equations of motion without anomalous terms.

In this appendix the relation between a Weyl semimetal Hamiltonian with spatially inhomogeneous tilt, and Weyl fermions in curved spacetimes is reviewed [17–27, 29]. The mapping at the level of the Hamiltonian can be established by identifying the tetrad corresponding to the metric with the effective tetrad in a Weyl semimetal [17, 18, 20, 26]. Here we take the simpler approach of demonstrating the equivalence at the level of the dispersion relation, specifically for a metric $g_{\mu\nu}$ with signature $(-+++)$ written in Painlevé-Gullstrand form. The four vector is denoted by $x^{\mu} = (ct, \boldsymbol{x})$ ($c$ is the speed of light). We are specifically interested in the Schwarzschild black hole metric written in Painlevé-Gullstrand coordinates $(t, \boldsymbol{r})$, which satisfies:

$$ds^2 = g_{\mu\nu}dx^{\mu}dx^{\nu} = -\left(1 - \frac{u(\boldsymbol{r})^2}{c^2}\right)c^2 dt^2 - 2\boldsymbol{u}(\boldsymbol{r}) \cdot d\boldsymbol{r}\, dt + d\boldsymbol{r}^2. \tag{A.1}$$

Here, $\boldsymbol{u}(\boldsymbol{r})$ has an interpretation of a local free-fall velocity field under the gravitation of a black hole. The matrix representation of the metric $g_{\mu\nu}$ can be identified as

$$g = \frac{1}{c}\begin{pmatrix} u(\boldsymbol{r})^2/c - c & -u_x(\boldsymbol{r}) & -u_y(\boldsymbol{r}) & -u_z(\boldsymbol{r}) \\ -u_x(\boldsymbol{r}) & c & 0 & 0 \\ -u_y(\boldsymbol{r}) & 0 & c & 0 \\ -u_z(\boldsymbol{r}) & 0 & 0 & c \end{pmatrix}. \tag{A.2}$$

For massless Weyl particles, the energy momentum relation is given by $g^{\mu\nu}k_{\mu}k_{\nu} = 0$, where $k_{\mu} = (-E/c, \boldsymbol{k})$ is the momentum four-vector and $g^{\mu\nu}$ is the inverse metric. For the acoustic metric, we find

$$0 = g^{\mu\nu}k_{\mu}k_{\nu} = k^2 - \frac{(E - \boldsymbol{u}(\boldsymbol{r}) \cdot \boldsymbol{k})^2}{c^2}. \tag{A.3}$$

The two solutions are the dispersion relation of a tilted Weyl cone

$$E_{\pm} = \boldsymbol{u}(\boldsymbol{r}) \cdot \boldsymbol{k} \pm c|\boldsymbol{k}|. \tag{A.4}$$

The tilt in the Weyl cone due to the nodal tilt profile is then interpreted as the tilt of the local lightcone structure at each point in a curved spacetime [17]. Expressed in spherical coordinates of the Schwarzschild spacetime, the velocity field reads explicitly $\boldsymbol{u}(\boldsymbol{r}) = -c\sqrt{r_H/r}\,\hat{\boldsymbol{r}}$, where $r_H = GM/c^2$ is the radius of the event horizon, $G$ denotes the universal gravitational constant and $M$ corresponds to the mass of a black hole. In electronic systems, the analogy is readily established by replacing $c \to v_F$, and a reinterpretation of $r_H \to r_t$, which marks the boundary between a type I and type II region where the Weyl cone is overtilted. In the main text, we consider tilt profiles with axial symmetry, but we argue below that the in-plane motion resulting from the geodesic equation is indeed identical if Berry curvature terms do not contribute to the semiclassical equations of motion.

While the above equivalence between dispersions was shown for a specific metric, we demonstrate the equivalence between Hamilton's equations and the geodesic equation, in the absence of electromagnetic fields and for massless particles. The Hamiltonian is given by [46]

$$\mathcal{H}(\{x^{\alpha}(\tau)\}, \{k_{\beta}(\tau)\}) = \frac{1}{2}g^{\mu\nu}(\{x^{\alpha}(\tau)\})k_{\mu}(\tau)k_{\nu}(\tau), \tag{A.5}$$

where $\{x^\mu(\tau)\}$ and $\{k_\mu(\tau)\}$ are the canonical variables for the coordinates and momenta, and we assume that the metric $g^{\mu\nu}$ depends only on the coordinates $\{x^\mu\}$. Hamilton's equations are

$$\frac{dx^\mu}{d\tau} = \frac{\partial \mathcal{H}}{\partial k_\mu} = g^{\mu\nu}k_\nu \,, \tag{A.6}$$

$$\frac{dk_\mu}{d\tau} = -\frac{\partial \mathcal{H}}{\partial x^\mu} = -\frac{1}{2}(\partial_\mu g^{\nu\sigma})k_\nu k_\sigma \,, \tag{A.7}$$

where we use the notation $\partial_\mu g^{\nu\sigma} = \frac{\partial g^{\nu\sigma}}{\partial x^\mu}$. For later convenience, the first equation is equivalent to

$$k_\mu = g_{\mu\nu}\frac{dx^\nu}{d\tau} \,. \tag{A.8}$$

The total derivative of the first equation reads

$$\frac{d^2 x^\mu}{d\tau^2} = g^{\mu\nu}\frac{dk_\nu}{d\tau} + (\partial_\lambda g^{\mu\nu})\frac{dx^\lambda}{d\tau}k_\nu = g^{\mu\nu}\frac{dk_\nu}{d\tau} + (\partial_\lambda g^{\mu\nu})\frac{dx^\lambda}{d\tau}g_{\nu\sigma}\frac{dx^\sigma}{d\tau} \,.$$

Using the second equation, we get

$$\frac{d^2 x^\mu}{d\tau^2} = -\frac{1}{2}g^{\mu\nu}(\partial_\nu g^{\sigma\lambda})k_\sigma k_\lambda + (\partial_\lambda g^{\mu\nu})g_{\nu\sigma}\frac{dx^\lambda}{d\tau}\frac{dx^\sigma}{d\tau} \,. \tag{A.9}$$

After inserting eq. (A.8), we arrive at

$$\frac{d^2 x^\mu}{d\tau^2} = -\frac{1}{2}g^{\mu\nu}(\partial_\nu g^{\sigma\lambda})g_{\sigma\alpha}g_{\lambda\beta}\frac{dx^\alpha}{d\tau}\frac{dx^\beta}{d\tau} + (\partial_\lambda g^{\mu\nu})g_{\nu\sigma}\frac{dx^\lambda}{d\tau}\frac{dx^\sigma}{d\tau} \,. \tag{A.10}$$

From $g_{\mu\nu}g^{\nu\sigma} = \delta_\mu^\sigma$ follows $\partial_\sigma g^{\mu\nu} = -g^{\mu\alpha}(\partial_\sigma g_{\alpha\beta})g^{\beta\nu}$, which leads to

$$\frac{d^2 x^\mu}{d\tau^2} = \frac{1}{2}g^{\mu\nu}g^{\sigma\gamma}g_{\sigma\alpha}(\partial_\nu g_{\gamma\kappa})g^{\kappa\lambda}g_{\lambda\beta}\frac{dx^\alpha}{d\tau}\frac{dx^\beta}{d\tau} - g^{\mu\alpha}(\partial_\lambda g_{\alpha\beta})g^{\beta\nu}g_{\nu\sigma}\frac{dx^\lambda}{d\tau}\frac{dx^\sigma}{d\tau} \tag{A.11a}$$

$$= \frac{1}{2}g^{\mu\nu}(\partial_\nu g_{\alpha\beta})\frac{dx^\alpha}{d\tau}\frac{dx^\beta}{d\tau} - g^{\mu\alpha}(\partial_\lambda g_{\alpha\sigma})\frac{dx^\lambda}{d\tau}\frac{dx^\sigma}{d\tau} \tag{A.11b}$$

$$= \frac{1}{2}g^{\mu\nu}\Big((\partial_\nu g_{\alpha\beta}) - 2(\partial_\alpha g_{\nu\beta})\Big)\frac{dx^\alpha}{d\tau}\frac{dx^\beta}{d\tau} \tag{A.11c}$$

$$= \frac{1}{2}g^{\mu\nu}\Big((\partial_\nu g_{\alpha\beta}) - (\partial_\alpha g_{\nu\beta}) - (\partial_\beta g_{\nu\alpha})\Big)\frac{dx^\alpha}{d\tau}\frac{dx^\beta}{d\tau} \tag{A.11d}$$

$$= -\Gamma^\mu_{\alpha\beta}\frac{dx^\alpha}{d\tau}\frac{dx^\beta}{d\tau} \,, \tag{A.11e}$$

where $\Gamma^\mu_{\alpha\beta} = g^{\mu\nu}(\partial_\alpha g_{\nu\beta} + \partial_\beta g_{\nu\alpha} - \partial_\nu g_{\alpha\beta})/2$ are the Christoffel symbols. This equation is directly the related to the acceleration, which is given as the covariant derivative

$$a^\mu(\tau) = \frac{dx^\alpha}{d\tau}\nabla_\alpha\frac{dx^\mu}{d\tau} = \frac{dx^\alpha}{d\tau}\left(\partial_\alpha\frac{dx^\mu}{d\tau} + \Gamma^\mu_{\alpha\beta}\frac{dx^\beta}{d\tau}\right) = \frac{d^2 x^\mu}{d\tau^2} + \Gamma^\mu_{\alpha\beta}\frac{dx^\alpha}{d\tau}\frac{dx^\beta}{d\tau} \,. \tag{A.12}$$

We therefore recognize that $(i)$ Hamilton's equations are equivalent to the geodesic equation and $(ii)$ curves which satisfy the geodesic equation are those with a vanishing acceleration $a^\mu(\tau) = 0$. To further establish a connection between Hamilton's equations and the semiclassical equations of motion, we need to change the parametrization from $\tau$ to $t$, which, as we demonstrate in the following, results in $\tau \to t$ and $\mathcal{H} \to E$ in Hamilton's equations for the spatial components of $x^\mu$ and $k_\mu$.

Assuming $g^{00} \neq 0$, the equation

$$\mathcal{H}\left(E, \{x^i\}, \{k_i\}\right) = \frac{E^2}{2c^2}g^{00} - \frac{E}{c}g^{0i}k_i + \frac{1}{2}g^{ij}k_i k_j = 0, \tag{A.13}$$

has two solutions for the energy $E(\{x^i\}, \{k_i\})$. From Hamilton's equations, we have

$$\frac{dt}{d\tau} = \frac{1}{c}\frac{dx^0}{d\tau} = \frac{1}{c}\frac{\partial\mathcal{H}}{\partial k_0} = -\frac{\partial\mathcal{H}}{\partial E}, \tag{A.14}$$

which contributes to the partial derivative of eq. (A.13) with respect to $k_i$, i.e.

$$0 = \frac{\partial\mathcal{H}}{\partial E}\frac{\partial E}{\partial k_i} + \frac{\partial\mathcal{H}}{\partial k_i} = -\frac{dt}{d\tau}\frac{\partial E}{\partial k_i} + \frac{\partial\mathcal{H}}{\partial k_i}. \tag{A.15}$$

Using eq. (A.6), we arrive at

$$\frac{dx^i}{d\tau} = \frac{\partial E}{\partial k_i}\frac{dt}{d\tau}, \tag{A.16}$$

in equivalence to the semiclassical equations of motion for the spatial components

$$\frac{dx^i}{dt} = \frac{dx^i}{d\tau}\left(\frac{dt}{d\tau}\right)^{-1} = \frac{\partial E}{\partial k_i}. \tag{A.17}$$

Similarly, from the partial derivative of eq. (A.13) with respect to $x^i$, we get

$$0 = -\frac{\partial E}{\partial k_i}\frac{dt}{d\tau} + \frac{\partial\mathcal{H}}{\partial x^i}, \tag{A.18}$$

resulting in the semiclassical equations of motion for the momenta

$$\frac{dk_i}{dt} = \frac{dk_i}{d\tau}\left(\frac{dt}{d\tau}\right)^{-1} = -\frac{\partial E}{\partial x^i}. \tag{A.19}$$

For a spherically symmetric tilt profile $\boldsymbol{u}(\boldsymbol{r}) = u(r)\hat{\boldsymbol{r}}$, the in-plane dynamics ($\theta = \pi/2$) is identical to the dynamics found for the axially symmetric case. This can be seen if the Hamiltonian in eq. (A.5) is expressed in spherical coordinates. The transformation is readily performed by $g \rightarrow J^T g J$, where $J$ is the Jacobian of the coordinate transformation $(x, y, z) \rightarrow (r\sin(\theta)\cos(\phi), r\sin(\theta)\sin(\phi), r\cos(\theta))$. We get

$$g \rightarrow \begin{pmatrix} u(r)^2/c^2 - 1 & -u(r)/c & 0 & 0 \\ -u(r)/c & 1 & 0 & 0 \\ 0 & 0 & r^2 & 0 \\ 0 & 0 & 0 & r^2\sin^2(\theta) \end{pmatrix}, \tag{A.20}$$

which results in

$$ds^2 \rightarrow -\left(1 - \frac{u^2(r)}{c^2}\right)c^2 dt^2 - 2u(r)dr dt + dr^2 + r^2(d\theta^2 + d\phi^2\sin^2(\theta)). \tag{A.21}$$

Using the inverse of the transformed metric, eq. (A.5) yields

$$\mathcal{H} \rightarrow \frac{1}{2}\left(k_1^2 + \frac{k_2^2}{r^2} + \frac{k_3^2}{r^2\sin(\theta)^2} - \frac{(E - k_1 u(r))^2}{c^2}\right), \tag{A.22}$$

where $k_i$ are the canonical momenta. They are related to $k_r$, $k_\theta$ and $k_\phi$ through the dispersion relation, the solutions of setting eq. (A.22) equal to zero, i.e.

$$E_\pm = k_1 u(r) \pm c \sqrt{k_1^2 + \left(k_2^2 + k_3^2 \csc^2(\theta)\right)/r^2} \,. \tag{A.23}$$

We identify $k_1 = k_r$ and $k_2^2 + k_3^2 \csc(\theta) = r^2(k_\theta^2 + k_\phi^2)$, where $k_r = \boldsymbol{k} \cdot \hat{\boldsymbol{r}}$, $k_\theta = \boldsymbol{k} \cdot \hat{\boldsymbol{\theta}}$, $k_\phi = \boldsymbol{k} \cdot \hat{\boldsymbol{\phi}}$. By definition, $k_x^2 + k_y^2 + k_z^2 = k_r^2 + k_\theta^2 + k_\phi^2$, and $\boldsymbol{u}(\boldsymbol{r}) \cdot \boldsymbol{k} = u(r)k_r$. Due to rotational symmetry, angular momentum is conserved. We impose $\boldsymbol{L} = L_z \hat{\boldsymbol{z}}$, with $L_z = rk_\phi$, which allows us to set $d\theta/d\tau = k_2/r^2 = 0$ (and therefore $k_2 = 0$) and thus constraints the motion onto the $x, y$ plane, i.e.

$$\frac{d\phi}{d\tau} = \frac{k_3}{r^2} \,, \tag{A.24}$$

$$\frac{dr}{d\tau} = k_r \left(1 - \frac{u(r)^2}{c^2}\right) + E\frac{u(r)}{c^2} = \pm\frac{1}{c}\sqrt{E^2 - \frac{k_3^2}{r^2}\left(c^2 - u(r)^2\right)} \,. \tag{A.25}$$

Furthermore, from Hamilton's equations we find that $k_3$ is a conserved quantity, and identify $k_3 = L_z$ as the angular momentum. Compared to eq. (8), note the absence of the factor $(dt/d\tau)^{-1} = (\partial \mathcal{H}/\partial E)^{-1} = \pm c/\sqrt{k_r^2 + L_z^2/r^2}$, which results in a dynamic effective mass when the equations are written in real time. Combining the above equations we get

$$\frac{d\phi}{dr} = \pm\frac{c\, L_z}{r^2\sqrt{E^2 - \varepsilon_{\mathrm{p,eff}}(r)}} \,, \tag{A.26}$$

which matches with eq. (10) after replacing $c \to v_F$ and $r \to \rho$. In spherically symmetric spacetimes, the separatrix always lies on a sphere, which in the context of black holes is known as the "photon sphere". For individual trajectories, the separatrices are great circles on the photon sphere. When we include Berry curvature effects, we find that the separatrix is still located on the photon sphere, but moves away from a great circle.

## B   Calculation of deflection angle for a power law tilt profile

In this appendix, we calculate the deflection angle for the case of lensing in the presence of a tilt profile of the form $u(\rho) = -v_F(\rho_t/\rho)^\alpha$. The deflection angle is to be calculated from

$$d\phi = \pm\frac{v_F L_z\, d\rho}{\rho^2\sqrt{E_+^2 - \varepsilon_{\mathrm{p,eff}}(\rho)}} \,. \tag{B.1}$$

In the following, the deflection angle will be expressed in terms of the tilt at the periapsis $\rho = \rho_p$, the point of closest approach to the center of the tilt profile. The tilt at the periapsis is given by $u_p^2 = u(\rho_p)^2 = v_F^2(\rho_t/\rho_p)^{2\alpha}$. The periapsis can be simply expressed in terms of the parameters $L_z$ and $E_+$, making use of the condition

$$\left.\frac{d\rho}{d\phi}\right|_{\rho=\rho_p} = 0 \,, \tag{B.2}$$

which implies that

$$0 = \left(\frac{d\rho}{d\phi}\right)^2\bigg|_{\rho=\rho_p} = \left(\frac{E_+}{L_z}\right)^2 \rho_p^4 - \rho_p^2\left(v_F^2 - u_p^2\right) \,, \tag{B.3}$$

and thus

$$\left(\frac{E_+}{L_z}\right)^2 = \frac{1}{\rho_p^2}\left(v_F^2 - u_p^2\right). \tag{B.4}$$

Therefore, the integral for the deflection angle $\varphi$ can be written as in eq. (11), namely,

$$\pi + \varphi = \int_{\phi_1}^{\phi_1 + \pi + \varphi} d\phi = \left(-\int_\infty^{\rho_p} + \int_{\rho_p}^\infty\right) \frac{v_F}{\sqrt{\frac{1}{\rho_p^2}\left(v_F^2 - u_p^2\right)\rho^4 - \rho^2\left(v_F^2 - u(\rho)^2\right)}} d\rho$$

$$= 2\int_{\rho_p}^\infty \frac{v_F}{\sqrt{\frac{1}{\rho_p^2}\left(v_F^2 - u_p^2\right)\rho^4 - \rho^2\left(v_F^2 - u(\rho)^2\right)}} d\rho, \tag{B.5}$$

where $\varphi = \phi_1 - \phi_2$ as shown in fig. 3. We can now expand the integrand for lensing trajectories. To leading order in $\rho_t/\rho_p \ll 1$, we find

$$\pi + \varphi \approx 2\int_{\rho_p}^\infty d\rho \left(\frac{\rho_p}{\rho\sqrt{\rho^2 - \rho_p^2}} + \frac{\rho_p\left(u_p^2\rho^2 - u^2\rho_p^2\right)}{v_F^2\rho\left(\rho^2 - \rho_p^2\right)^{3/2}}\right). \tag{B.6}$$

The first term gives exactly $\pi$. For the second term, we change the integration variable to the dimensionless $s = \rho/\rho_p$ to obtain the final expression for the lensing angle

$$\varphi \approx \frac{u_p^2}{v_F^2}\int_1^\infty ds \frac{s^2 - s^{-2\alpha}}{s\left(s^2 - 1\right)^{3/2}} = \frac{u_p^2}{v_F^2}\frac{\sqrt{\pi}\,\Gamma\left(\frac{3}{2} + \alpha\right)}{\Gamma(1 + \alpha)}, \tag{B.7}$$

valid for $\rho_t/\rho_p \ll 1$.

## C  Calculation of the transverse shift for a power law tilt profile

In this appendix, we compute the transverse shift $\Delta z$ for $k_z = 0$ (as fixed throughout the main text). The starting point are eq. (6) and eq. (7a), from which we obtain

$$\frac{dz}{d\rho} = \frac{\dot{z}}{\dot{\rho}} = -\frac{\chi}{2k^3}\frac{\frac{\boldsymbol{k}\cdot\hat{\boldsymbol{\rho}}}{\rho}\left(\frac{\partial u}{\partial \rho} - \frac{u}{\rho}\right)L_z}{u + v_F\frac{\boldsymbol{k}\cdot\hat{\boldsymbol{\rho}}}{k}}. \tag{C.1}$$

To make progress, we furthermore use

$$E_+ = u\boldsymbol{k}\cdot\hat{\boldsymbol{\rho}} + v_F k, \tag{C.2}$$

$$k^2 = (\boldsymbol{k}\cdot\hat{\boldsymbol{\rho}})^2 + L_z^2/\rho^2, \tag{C.3}$$

to express $\boldsymbol{k}\cdot\hat{\boldsymbol{\rho}}$ and $k$ in terms of the conserved quantities $E_+$ and $L_z$.

We find the pairs of solutions $(\boldsymbol{k}\cdot\hat{\boldsymbol{\rho}}, k) \to (k_\rho^{(\sigma)}, k^{(\sigma)})$ with $\sigma = \pm 1$ and

$$k_\rho^{(\sigma)} = \frac{E_+ u - \sigma\frac{\sqrt{L_z^2 u^2 v_F^2 - L_z^2 v_F^4 + E_+^2 v_F^2\rho^2}}{\rho}}{u^2 - v_F^2}, \tag{C.4}$$

$$k^{(\sigma)} = \sqrt{\left(k_\rho^{(\sigma)}\right)^2 + \frac{L_z^2}{\rho^2}}. \tag{C.5}$$

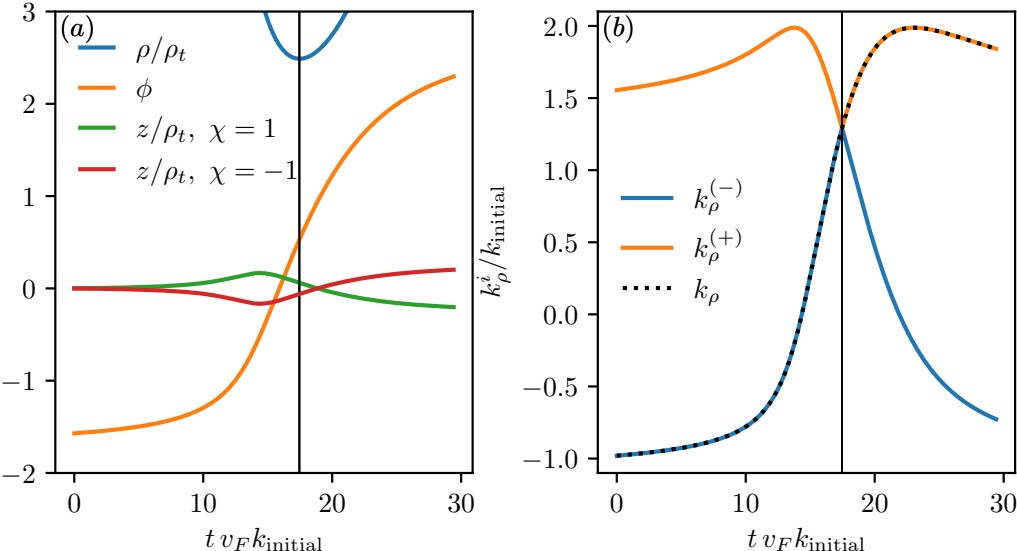

Figure 12: Coordinates of an arbitrary lensing trajectory. Spatial coordinates are plotted in panel ($a$), while panel ($b$) displays the two roots $k_\rho^{(\sigma)}$, together with $k_\rho$ of the trajectory. At the periapsis (marked by a black line), $k_\rho$ switches between the two branches.

The two signs of $\sigma$ define the expressions of $k_\rho^{(\sigma)}$ and $k^{(\sigma)}$, while approaching ($\sigma = +1$) and escaping ($\sigma = -1$) from the strongly tilted region (see fig. 12). Recalling that the point of closest approach for a lensed trajectory is the periapsis $\rho_p$, the total transverse shift is given by

$$
\begin{aligned}
\Delta z &= -\int_\infty^{\rho_p} d\rho\, \frac{\chi}{2\,(k^{(-)})^3} \frac{\frac{k_\rho^{(-)}}{\rho}\left(\frac{\partial u}{\partial \rho} - \frac{u}{\rho}\right)L_z}{u + v_F\,\frac{k_\rho^{(-)}}{k^{(-)}}} - \int_{\rho_p}^\infty d\rho\, \frac{\chi}{2\,(k^{(+)})^3} \frac{\frac{k_\rho^{(+)}}{\rho}\left(\frac{\partial u}{\partial \rho} - \frac{u}{\rho}\right)L_z}{u + v_F\,\frac{k_\rho^{(+)}}{k^{(+)}}} \\
&= \int_{\rho_p}^\infty d\rho \left( \frac{\chi}{2\,(k^{(-)})^3} \frac{\frac{k_\rho^{(-)}}{\rho}\left(\frac{\partial u}{\partial \rho} - \frac{u}{\rho}\right)L_z}{u + v_F\,\frac{k_\rho^{(-)}}{k^{(-)}}} - \frac{\chi}{2\,(k^{(+)})^3} \frac{\frac{k_\rho^{(+)}}{\rho}\left(\frac{\partial u}{\partial \rho} - \frac{u}{\rho}\right)L_z}{u + v_F\,\frac{k_\rho^{(+)}}{k^{(+)}}} \right).
\end{aligned}
\tag{C.6}
$$

Next, we use that $u(\rho) = -v_F\,(\rho_t/\rho)^\alpha$, as well as $E_+ = \frac{|L_z|}{\rho_p}\sqrt{v_F^2 - u_p^2}$ [see eq. (B.4)]. We furthermore expand the integrand for lensed trajectories that stay far from the tilt center, i.e., trajectories for which $\rho_t/\rho \ll 1$, which implies $u, u_p \ll v_F$. To leading order, the integral can then be approximated as

$$
\Delta z \approx \chi \int_{\rho_p}^\infty d\rho\, \frac{u^2\,(1+\alpha)\,\rho_p^2\,(\rho^2 - 2\rho_p^2)}{L_z\, v_F^2\, \rho^3\, \sqrt{\rho^2 - \rho_p^2}}.
\tag{C.7}
$$

It is finally helpful to bring the integration variable to a dimensionless form by setting $s = \rho/\rho_p$, which yields

$$\Delta z \approx \chi \, \frac{(1+\alpha)}{L_z} \, \rho_p \left(\frac{\rho_t}{\rho_p}\right)^{2\alpha} \int_1^\infty ds \, \frac{s^{-3-2\alpha}\left(s^2-2\right)}{\sqrt{s^2-1}}$$
$$= -\chi \, \frac{(1+\alpha)}{L_z} \, \rho_p \left(\frac{\rho_t}{\rho_p}\right)^{2\alpha} \frac{\sqrt{\pi}\,\alpha}{2} \frac{\Gamma\left(\alpha+\frac{1}{2}\right)}{\Gamma(2+\alpha)}$$
$$= -\chi \, \frac{u_p^2}{v_F^2} \frac{\rho_p}{L_z} \frac{\sqrt{\pi}\,\Gamma\left(\alpha+\frac{1}{2}\right)}{2\,\Gamma(\alpha)}, \qquad\qquad \text{(C.8)}$$

valid for $\rho_t/\rho_p \ll 1$.

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
