# Peer review of "Black hole mirages: electron lensing and Berry curvature effects in inhomogeneously tilted Weyl semimetals"

_SciPost Physics, doi:SciPost Phys. 14, 119 (2023)_

## Round 2 · Referee Report · Jasper van Wezel (Referee 1) · 2022-12-2

Strengths

1 well-written
2 thorough analysis on multiple scales
3 novel and timely
4 opens up an area of “tilt-tronics”.

Weaknesses

1 precise connection to gravity and black hole physics unclear (see below)
2 in parts: re-branding known physics (see below)

Report

The authors discuss electron transport in Weyl semimetals with an inhomogeneous tilt profile of the Weyl cones. The discussion is timely, generalising several recent works on the emergence of analogue black holes in inhomogeneously tilted Weyl semimetals, and applying these to the practically relevant setting of two and three dimensional Weyl semimetals. Finally, it addresses the electron transport in various relevant regimes, including those with non-trivial Berry curvature effects, and it makes a case for the possibility of extending the current approach to the development of tilt-tronics.

All of this renders the work relevant and interesting to a broad community of physicists. The paper is also clear, self-contained, and well-written. Based on this, I do believe the current work warrants publication in SciPost Physics. However, there are a few points in the discussion that remained unclear to me, which I would like to ask the authors to address:

1) Although the presented work is motivated by previous work on analogue black holes, I do not see any direct use of analogue gravity in the present paper. The tilt profiles used by the authors are inspired by black holes, but no black hole physics is discussed in the present work, the tilt profiles and state evolution are not matched to gravitational physics (see also below), and the majority of the analysis focusses on relatively high energies, where the correspondence to any gravitational system breaks down. Altogether, the paper reads like a thorough and useful exposition of tilt-tronics and trajectory-engineering, but the link to gravitational physics appears tenuous.

2) The correspondence between the specific tilt profile that is the main focus of the present work and a gravitational black hole, is unclear in several ways: — the trajectories presented in fig 1 and the effective potential presented in fig 2 contain large regions in which the semiclassical particle is repelled from the origin. Clearly this does not correspond to a situation with a (universally attractive) gravitational mass concentrated at the origin. — the effective potential depicted in fig 2 does not contain a singularity at any point. — the correspondence of the semiclassical trajectories to geodesic solutions of any gravitational metric is never made. Appendix A shows a recipe for finding such a correspondence in general, but does not reproduce the semiclassical trajectories in the main text (including ones accelerating away from the black hole) starting from a metric and using Einstein’s equations. — No horizon is identified in the presented semiclassical dynamics or potential.

3) It is not made clear in the present text how the (semiclassical and fully quantum) trajectories connect to the specific tilt profile. As the authors explain, the “horizon” in their construction coincides with the point where a type-I Weyl semimetal transitions into a type-II Weyl semimetal. At this point, the group velocity in the Weyl cone dispersion along the radial direction is zero. The fact that semiclassical trajectories transition through the horizon, and that fully quantum trajectories can even exit the horizon from the inside out, are in apparent contradiction with the naive observation of zero velocity indicating a black hole horizon. Clearly both types of behaviour must be caused by a combination of band curvature and scattering events. How this “high energy” physics enters the effective description of the (semiclassical or fully quantum) trajectories is not discussed in the present manuscript.

4) I find the discussion of a “mirage” appearing in the title and several places in the manuscript very misleading. The low-energy effective description that is argued to mimic gravitational dynamics is found to break down upon approaching the horizon. All this means however, is that the low-energy description breaks down when it is probed at short length scales or high energies. This is not surprising at all, as the low-energy description was only ever an emergent (effective) description. Saying that the emergent long-wavelength physics is a “mirage” because it disappears when probed at short (lattice) length scales, is like saying that a table is not really rigid because its rigidity disappears when probed at the atomic scale. I think this is an unfair rebranding of the already well-known limits of emergence.

5) It is unclear at the moment why trajectories in the simulations stop at the origin, or why in section 3.2 it is mentioned that wave pakets should accumulate at the origin. From the semiclassical dynamics, the trajectories reach the origin with non-zero velocity, and there is thus no reason for them to suddenly stop there, rather than continuing, bouncing, or scattering. In the fully quantum dynamics, wave packet evolution is unitary, and accumulation at the origin is fundamentally disallowed.

6) minor point: figure 4 appears to have a wrong title on the bottom right graph.

Requested changes

I suggest the authors either include additional details and discussion addressing the points above, or rephrase some of the claims they make in order to avoid these issues.

  • validity: high
  • significance: high
  • originality: high
  • clarity: top
  • formatting: perfect
  • grammar: perfect

Author:  Andreas Haller  on 2023-02-01  [id 3297]

(in reply to Report 1 by Jasper van Wezel on 2022-12-02)

Dear Referee,

we thank you for your valuable suggestions and criticism, which allowed us to improve our manuscript.
Please find in the attached PDF our detailed reply to your report.

Kind regards,
On behalf of all authors,
Andreas Haller

Attachment:

reports_reply_32SMuIl.pdf

---

## Round 2 · Referee Report · Teemu Ojanen (Referee 2) · 2022-12-12

Strengths

  1. Clear presentation
  2. Systematic comparison of the different effective theories
  3. Insightful classification of different kinetic regimes

Weaknesses

  1. Optimistic assumptions regarding solid-state materials

Report

The authors consider Weyl materials with a spatially-varying tilt profiles and make connection to the particle kinetics near a black hole. This connection arises due to the presence of tilted Weyl cones in the effective long-wavelength theory. Studies similar in spirit are numerous, if one also counts studies in 2d materials. However, the present paper makes a number of significant contributions to the field of emergent curved-space physics in Weyl and Dirac materials.

The authors study particle kinetics in three increasingly realistic frameworks: the long-wavelength semiclassics, the lattice-regulated semiclassics and the fully microscopic scattering theory. I am not aware of previous works that would systematically cover all the three levels of description. The long-wavelength theory is particularly insightful regarding the curved-space kinetics, which is essential hidden in more accurate formulations. Here the paper makes its first important contribution in identifying different length scales in the problem of an axisymmetric "black hole" where the event horizon is marked by the interface between the 1st and 2nd type of Weyl dispersions. The equations of motion at this level predict the existence of separatrix which distinguishes the escaping and in-falling trajectories. While the conclusions of the linearized semiclassics cannot be taken as a face value, the authors go on to argue their meaning in more accurate descriptions. Working with a particular lattice model, the authors write down the semiclassical equations and solve them. This no longer admits analytical progress, and the connection to curved-space kinetics is obscured. However, the resulting trajectories are very well in agreement with the long-wavelength theory where expected, and indicate where the long-wavelength theory is no longer applicable. Finally, the predictions of semiclassics are compared to fully quantum-mechanical scattering calculations by comparing the trajectories of the former to the electron density obtained from the latter. While this is limited to 2d motion, this is very exciting since I have not seen similar comparison before. It lends a lot of credibility to the semiclassical treatments and qualitatively motivates a large number of previous works employing such methods. It also allows the authors to pinpoint the breakdown of the semiclassics in the regime where the gradient suggests an event horizon. Granted that the semiclassics cannot accurately describe extreme textures, it still seems as surprisingly accurate tool to understand the particle kinetics qualitatively. I think that the painstaking comparison of the different descriptions and justification of the semiclassics away from the overtilt regime is the main contribution of this work.

I do not find much to criticise in this work. However, there are some looming questions in this field that applies to all the studies. While it does not invalidate the results on the manuscript in anyway, one still feels a bit pessimistic about realizing the required textures in solid-state materials. I do not doubt that the discussed effects can be observed in highly idealized quantum simulators but arranging the needed strain or doping textures in solids seems difficult. Moreover, there are always some concentration of defects and artificial material manipulations are likely to increase them. Thus, the purely ballistic particle motion seems improbable. Of course, theorists need to start somewhere and including all the real-world complications is not a logical starting place. Still, acknowledging their problematic nature prominently is a good idea. I can imagine how much of the discussed physics would survive even in the presence of impurities, so I share the overall optimism regarding these studies.

Requested changes

Rather than requested changes, I would propose a short discussion of the effects of impurity scattering and the related hierarchy of length scales that the system should satisfy in order to preserve/wipe out the studied effects.

  • validity: high
  • significance: high
  • originality: high
  • clarity: high
  • formatting: good
  • grammar: excellent

Author:  Andreas Haller  on 2023-02-01  [id 3296]

(in reply to Report 2 by Teemu Ojanen on 2022-12-12)

Dear Referee,

we thank you for your valuable suggestions and criticism, which allowed us to improve our manuscript.
Please find in the attached PDF our detailed reply to your report.

Kind regards,
On behalf of all authors,
Andreas Haller

Attachment:

reports_reply_0e9FqHn.pdf

---

## Round 2 · Referee Report · Anonymous (Referee 3) · 2022-12-16

Strengths

  1. Comprehensive analysis of when the analogy between black hole and space-dependent tilt breaks down.
  2. Unlike other theoretical works, this work features a comparison between semiclassical treatment and tight-binding calculations.
  3. Explicit analysis of the regime of validity of semiclassics.
  4. Discussion of chirality depending trajectories which can lead to interesting chiral separation effects.
  5. Timely subject: one the one hand lensing and focusing Dirac and Weyl fermions has a renewed interest as devices get better. On the other hand, gravitational analogues are being pushed.

Weaknesses

  1. The focus is largely towards semiclassics: the reasons tight binding deviates from semiclassics are discussed only superficially and most are left for future work. Admittedly, these are often challenging to compare.
  2. I lacked a bit more context and connection to other works, particularly those discussing anomalous effects. I touch upon this point further in my report.
  3. While applicable to other papers on the topic as well, it is hard to justify how this simple picture might be realizable in a real 3D solid-state material.

Report

The paper is timely and clearly written. It provides a link between lensing effects more familiar in the context of black holes and that of space dependent parameters of Weyl semimetals. In my opinion, one a key contribution of this paper is to establish where these appealing analogies break down, and does not oversell the results. Specifically, I really appreciate the comparison to tight-binding simulations, even if the origin of the deviations are not entirely well understood.

The results are sound and the analysis shows transparent and explicit connections between the mentioned fields, with an honest analysis of their regime of applicability. The choice and description of the title in the main text are appropriate. Therefore I am willing to recommend publication. There are a few points that I would like the authors to discuss further before doing so:

  1. It is not clear to me what specific phenomenon would a space dependent tilt result in that would not be possible with a space dependent fermi velocity or Weyl node separation. As the authors mention, there are previous papers that use a curved metric to interpret the results (e.g. ref. 18). I believe the manuscript would benefit from a discussion of the differentiating elements of the lenses discussed in previous works and those discussed here. In particular, is there any differentiating experiment (even if it is a gedanken experiment) when one could differentiate curved trajectories coming from space dependent tilts versus space dependent Weyl node separation?

  2. Additionally, I found interesting that a tilt variation combined with the anomalous (Berry curvature) term gives rise to a chiral separation effect, where one chirality moves upwards and the other one downwards (as shown in Fig. 10). This behaviour is very reminiscent of anomalous currents that distinguish left and right chiralities. Can the authors link these results with anomalous currents? For example there are frame effects that can lead to seemingly anomalous currents.

Requested changes

Connected to the above points I would encourage the authors to:

  1. Contextualize further their lensing results from previous lensing proposals.
  2. Discuss the differences between lensing arising from Weyl node separation variation versus tilt variation.
  3. Discuss the connection with anomalous currents, if possible.

  • validity: high
  • significance: high
  • originality: high
  • clarity: top
  • formatting: perfect
  • grammar: perfect

Author:  Andreas Haller  on 2023-02-01  [id 3295]

(in reply to Report 3 on 2022-12-16)

Dear Referee,

we thank you for your valuable suggestions and criticism, which allowed us to improve our manuscript.
Please find in the attached PDF our detailed reply to your report.

Kind regards,
On behalf of all authors,
Andreas Haller

Attachment:

reports_reply.pdf

---

## Round 3 · Referee Report · Jasper van Wezel (Referee 1) · 2023-2-7

Strengths

1 well-written
2 thorough analysis on multiple scales
3 novel and timely
4 opens up an area of “tilt-tronics”.

Weaknesses

Precise connection to gravity and black hole physics remains unclear on one point (see below).

Report

I would like to thank the authors for their careful and thorough consideration of my earlier remarks.

The responses to questions 1.2, 1.3, 1.4, and 1.5 are convincing, and I thank the authors for including several clarifications and a more nuanced discussion of the "mirage" effect in their manuscript.

I am not really convinced by the response of the authors to question 1.1, however.

In particular, the reference mentioned by the authors in 1.1.1 focusses on a so-called Kiselev black hole (including "quintessence") rather than a Schwarzschild black hole. In passing, it does also show the effective potential for the pure Schwarzschild black hole (their eq 14), but that result in fact clearly shows that V_eff is proportional to the angular momentum L^2. For radial geodesics such as the red line in Figure 1 of the current manuscript, the effective potential is negative everywhere and corresponds to the uniform attraction expected for a Schwarzschild black hole (as shown more explicitly for example in figure 3 of arxiv:1911.04305). This contradicts the outward acceleration of the radial red line shown in figure 1 of the current manuscript.
I am therefore not convinced the situation described by the authors really does represent a Schwarzschild black hole, and I encourage the authors to clarify this inconsistency.

Notice also that the caption of Figure 2 in the current manuscript does not indicate which value for the angular momentum is used to generate the curves in the left panel.

Furthermore, in appendix A the authors derive geodesics in a Schwarzschild metric, and they show that the tilt profile for Weyl cones considered in their manuscript matches the local tilts of light cones in the Painlevee-Gullstrand coordinate description of the Schwarzschild metric. However, they do not show that the worldlines of semiclassical wavepackets subjected to their specific Weyl cone tilt profile follow the geodesics of freefalling particles in the gravitational metric.
My reason for asking the authors to explicitly demonstrate this, is that the outward radial acceleration of the red line in fig 1 does not seem to agree with the naive expectation for geodesics in the gravitational setting (see also 1.1.1). If there was no such conflicting observation, I would agree with the authors that further elaboration on the correspondence is superfluous. As it stands, however, I do not see how the results of appendix A agree with the red trajectory shown in Fig 1.

Please note that the observation of trajectories escaping the black hole in Sec 4 (which clearly do not correspond to gravitational black hole geodesics) is sufficiently explained in the revised manuscript by the added discussion of the effects of lattice dispersion.

If the authors can address this final point of confusion, I am happy to recommend publication in SciPost Physics.

Requested changes

I suggest the authors address the one remaining point of confusion identified above.

  • validity: high
  • significance: high
  • originality: high
  • clarity: top
  • formatting: perfect
  • grammar: perfect

Author:  Andreas Haller  on 2023-02-17  [id 3372]

(in reply to Report 1 by Jasper van Wezel on 2023-02-07)

We are indebted to Mr. van Wezel for being persistent on his last doubts. Before we submit a revised version, we would like to eliminate the last point of confusion.

We believe the main reason for his doubt is rooted in the fact that radial trajectories are accelerated during their outwards movement, such that we are focussing on this discussion in more detail here. Consider initial coordinates $x>\rho_t$, $y=z=0$ and $k_x>0$, $k_y=k_z=0$, such that the radial trajectory is along the $x$-axis. The equations of motion can then be solved for the momentum coordinate
$$
k_x(t)=k_x(0)^2\exp\left(-\int_0^t u'(x(\tau))d\tau\right).
$$
From the solution of $k_x$, we recognize $\dot x(t) = u(x) + v_F>0$. One can also show that the acceleration is $\ddot x(t) = u'(x)(u(x) + v_F)>0$. Also, since $L_z=0$, the effective potential vanishes, and $\dot x^2(t) = E_+^2/k^2=E_+^2\exp(2\int_0^t u'(x(\tau))d\tau)/k_x(0)^2$.

We want to stress that the radial velocity and the effective potential in our manuscript are consistent with Eqs. (12), (13) [for $\sigma=0$] and Fig. 1 of Phys. Rev. D 92, 084042. The quantitative differences in the amplitude vs. distance of Fig. 1 compared to our Fig. 2 are caused by choosing dimensionless units $GM=c=1$, which means $r_t/2=v_F=1$ in our setting. The crucial difference to the dynamics we present in the main text is that we parametrize the differential equations using the temporal coordinate, which differs with respect to Eq. (12) by an additional factor $({dt}/{d\tau})^{-1}$ (for details, we refer to the revised App. A). Note that this factor is equal to $\pm c/k$ when $L=L_z$, which gives rise to a decreasing effective mass for radially escaping trajectories,
$$
m_{\rm eff}(t)=2k^2=2k(0)^2\exp\left(-2\int_0^tu'(\rho(\tau))d\tau\right),
$$
and the particle is accelerated away from the origin.

The geodesic equation is parametrized by an affine parameter instead, resulting in $\dot \rho(\tau)^2=\pm E_\pm^2$ (for radial trajectories with $L_z=0$), such that the velocity of particles escaping radially is constant in $\tau$. In conclusion, the "repulsion" of radially escaping particles is caused by the effective mass as a result from the temporal coordinate parametrization of the trajectories, not by the effective potential.

Compared to arXiv:1911.04305v1 mentioned by the Referee, note that the authors use a different notation, i.e. $h^2=L^2/(m\mu)$ and $E=kmc^2$ (see also the Wikipedia article on Schwarzschild geodesics, and our reply to the second report). Therefore, their effective potential in Eq. (3.21) is written in different units, which, upon multiplying the right hand side of Eq. (3.21) by $m$, followed by $m\rightarrow0,\mu\rightarrow1$ leads to the equations in our main text.

We admit that we could have given more details in App. A relating the geodesic equation parametrized by an affine parameter to the semiclassical equations of motion parametrized by the temporal coordinate, which we will incorporate if Mr. van Wezel agrees to the conclusion of this discussion.

Jasper van Wezel  on 2023-02-17  [id 3373]

(in reply to Andreas Haller on 2023-02-17 [id 3372])

Dear authors,

thank you very much for reaching out and for the detailed arguments.
I think a clarification of appendix A using affine parameters and the e.o.m. using temporal coordinates will be very useful.

My confusion however, is much more basic.

If you have x(0)>0 and d^2 x/dt^2 x>0, then the particle is accelerating away from the origin.
If at the same time you claim that this particle trajectory coincides with a geodesic found in Schwarzschild spacetime, then that means there needs to be a free-falling observer in the Schwarzschild spacetime who is accelerated away from the black hole.
I don't understand how that could be the case, as all (zero angular momentum) free-fall motion should be attracted towards the black hole, not repelled away from it.

If you could please clarify this one point, that would be great.

Anonymous on 2023-02-20  [id 3382]

(in reply to Jasper van Wezel on 2023-02-17 [id 3373])

Dear Mr. van Wezel,

We sincerely thank you for your question - digging into it has led us to some very enjoyable reading. On a basic level, it is straightforward to see that the velocity of an outwards particle, written in Gullstrand-Painlevé coordinates does feature a positive acceleration, if measured in coordinate time, i.e. a "repulsion". For a quick and dirty check, see this link. To add a bit of physics, consider a particle falling into a black hole. As you’re certainly aware, a standard textbook view is that an observer asymptotically far away from the black hole will see the particle become slower and slower as it approaches the black hole - gravity thus seems to decelerate or "repel". On a deeper level, we found PhysRevD.25.3191 and in particular the conclusion of arXiv:1606.08300v2 helpful. Overall, your question is very well posed, and as you can see reveals that GR is simply fun to work with.

On behalf of all authors Kind regards, Andreas Haller

---

## Round 3 · Referee Report · Anonymous (Referee 3) · 2023-2-10

Report

I would like to thank the authors for their concise remarks on my comments.
Specifically thanks to their answer of point 3.2 it is now clearer what differences can be expected between a space dependent tilt and a space-dependent Fermi velocity.

Regarding point 3.1 it is true that they mention several references in the introduction. The goal of my comment was not to add more references, but rather elaborate further on what concretely differentiates their predictions from earlier works. Thanks to their reply to point 3.2 it is possible to guess somewhat the differences and I will not insist more on this comment.

Their reply to 3.3 is rather short and not very concrete, but I guess this attests to the difficulty of the question. Since I raised it as optional, I won't dwell either on it.

Therefore, I recommend publication in SciPost.
  • validity: high
  • significance: good
  • originality: good
  • clarity: top
  • formatting: perfect
  • grammar: perfect

Author:  Andreas Haller  on 2023-02-17  [id 3370]

(in reply to Report 2 on 2023-02-10)

We thank the Referee for his/her report, and like to add the following remarks on point 3.1 and 3.2. In Schwarzschild coordinates, the matrix representation of the metric $g_{\mu\nu}$ reads
\begin{align}
g =
\begin{pmatrix}
1-\frac{u(r)^2}{c^2} & 0 & 0 & 0 \\
0 & -\frac{1}{1-\frac{u(r)^2}{c^2}} & 0 & 0 \\
0 & 0 & -r^2 & 0 \\
0 & 0 & 0 & -r^2 \sin^2(\theta )
\end{pmatrix}.
\end{align}
This leads to the Hamiltonian
\begin{align}
\mathcal H =
\frac{1}{2} \left(\frac{k_1^2 u(r)^2}{c^2}+\frac{E^2}{c^2-u(r)^2}-k_1^2-\frac{k_2^2}{r^2}-\frac{k_3^2 \csc^2(\theta )}{r^2}\right),
\end{align}
with a dispersion relation on the $(x,y)$ plane, assuming $r>r_t$
\begin{align}
E_\pm = \pm c\sqrt{1-\frac{u(r)^2}{c^2}}\sqrt{k_1^2+\frac{L_z^2}{r^2}}
\end{align}
and therefore a different dynamics parametrized by the temporal coordinate. We note that the analog setup could be naively realized by a spatially modulated Weyl cone of the form $E_\pm=\pm v(r) k$ (using $k_z=0$, like in our manuscript, such that contributions to the dynamics by anomalous terms vanish). The associated Hamiltonian is given by $\mathcal H = \chi\,v(r)\,{\vec k}\cdot {\vec\sigma}$, where the tilt is translated into an isotropic velocity field $v(r)=\sqrt{c^2-u(r)^2}$. Note that, contrary to Gullstrand-Painlevé coordinates, Schwarzschild coordinates only make sense for $r>r_t$ because the metric is divergent at $r=r_t$, and the dispersion becomes complex at $r<r_t$. Despite these differences, we find by straightforward evaluation that the effective potential identified from $\dot r(\tau)$ for $r>r_t$ is equal in both coordinate systems.

---

## Round 3 · Author Response

Dear Editor,

Thank you for sending us the reports of the Referees. We are very grateful for their careful, positive and constructive reports that helped us improve the manuscript. We have added technical and logical steps and clarified some parts in the new version of the paper, complying with the requests of all Referees. Please find our more detailed reply in the sections below. In view of the already positive reports in the last round, we hope that our work can now be accepted for publication on SciPost Physics.

On behalf of all authors,
Kind regards,
Andreas Haller

---

## Round 3 · List of Changes

Sec. 1 and 2: - Included a paragraph on the discussion of the effective potential - Added references to the introduction

Sec. 3: - Avoided the notion of "accumulation at the tilt center".

Sec. 4: - Extended and revised Sec. 4.1 and 4.3

Sec. 5: - Added paragraph on "anomalous currents"

Conclusion: - Added conclusion from the new extension of Sec. 4

---

## Round 4 · Referee Report · Jasper van Wezel (Referee 1) · 2023-2-22

Strengths

1 well-written
2 thorough analysis on multiple scales
3 novel and timely
4 opens up an area of “tilt-tronics”

Report

I would like to sincerely thank the authors for their patience and careful consideration of my comments.
The latest reply and resubmission resolve my remaining questions.

My earlier confusion about Figure 1 stemmed from the fact that the red line, and only that line, concerns an outgoing light ray, corresponding to a wave packet on the opposite branch of the Weyl node as compared to all other trajectories in the figure.
The outgoing mode, however, is still being described with the proper time belonging to infalling observers. The infalling observer sees the outgoing light frozen at the horizon and then speed up as it escapes to infinity, as explained by the authors.

I did not notice before that the red line was different from all other lines in the figure, because the entire section 3.0 is about defining different types of infalling modes (lensing, capture, and seperatrix), and not outgoing ones. In the latest revision, the authors added a line at the end of this secion mentioning the difference between the red line and the others. I would (optionally) suggest that adding a similar comment already in the caption of figure 1 could be very helpful to readers.

I am happy to now recommend publication of this work in SciPost Physics.

---

## Round 4 · Author Response

We thank both Referees for their useful suggestions, which allowed us to improve our manuscript.

In particular, Appendix A of the revised version contains more details on the derivation of the semiclassical equations of motion. We further fixed an issue with the effective mass of trajectories (it was defined as an inverse mass in the earlier versions).

---

## Round 4 · List of Changes

• Revised appendix A
  • Changed the definition of the effective mass

---

## Editorial Decision

published